# The Advantage of Conditional Meta-Learning for Biased Regularization and Fine Tuning

**Giulia Denevi** [1,*]    **Massimiliano Pontil** [1,2]    **Carlo Ciliberto** [3]

[1] University College of London (UK)

[2] Istituto Italiano di Tecnologia (Italy)

[3] Imperial College of London (UK)

[*] Work done while the first author was with Istituto Italiano di Tecnologia (Italy)

*g.denevi@ucl.ac.uk, massimiliano.pontil@iit.it, c.ciliberto@imperial.ac.uk*

## Abstract

Biased regularization and fine tuning are two recent meta-learning approaches. They have been shown to be effective to tackle distributions of tasks, in which the tasks' target vectors are all close to a common meta-parameter vector. However, these methods may perform poorly on heterogeneous environments of tasks, where the complexity of the tasks' distribution cannot be captured by a single meta-parameter vector. We address this limitation by conditional meta-learning, inferring a conditioning function mapping task's side information into a meta-parameter vector that is appropriate for that task at hand. We characterize properties of the environment under which the conditional approach brings a substantial advantage over standard meta-learning and we highlight examples of environments, such as those with multiple clusters, satisfying these properties. We then propose a convex meta-algorithm providing a comparable advantage also in practice. Numerical experiments confirm our theoretical findings.

## 1   Introduction

Biased regularization and fine tuning [5, 13, 14, 16, 17, 19, 23, 24, 28, 31, 33] are two recent optimization-based meta-learning techniques that transfer knowledge across an environment of tasks by leveraging a common meta-parameter vector. Their origin and inspiration go back to multi-task and transfer learning methods [10, 15, 27], designed to address a prescribed set of tasks with low variance. These techniques can be described as a nested optimization scheme: while at the within-task level, an inner algorithm performs tasks' specific optimization with the current meta-parameter vector, at the meta-level a meta-algorithm updates the aforementioned meta-parameter by leveraging the experience accumulated from the tasks observed so far. In biased regularization the inner algorithm is given by the within-task regularized empirical risk minimizer and the meta-parameter vector plays the role of a bias in the regularizer, while fine tuning employs online gradient descent as the within-task algorithm and the meta-parameter vector is the associated starting point.

Despite their success, the above methods may fail to adapt to heterogenous environments of tasks, in which the complexity of the tasks' distribution cannot be captured by a single meta-parameter vector. In literature, a variety of methods have tried to address this limitation by clustering the tasks and, then, leveraging tasks' similarities within each cluster [2, 4, 21, 29, 30]. However, such methods usually lead to non-convex formulations [2, 4] or provide only partial guarantees on surrogate convex problems [21, 30]. As alternative, recent approaches in meta-learning literature advocated learning a conditioning function that maps a task's dataset into a meta-parameter vector that is appropriate for the task at hand [9, 22, 35, 39–41]. This perspective has been shown to be promising in applications, however theoretical investigations are still lacking. The authors of [40] have recently made a first

step towards this theoretical investigation adopting a structured prediction perspective for conditional meta-learning. However, their method relies on a non-convex formulation of the problem. This allows them to provide guarantees only on the generalization properties of their method, but not on its estimation properties. In this work, we address the limitation above for biased regularization and fine tuning by developing a new conditional meta-learning framework. Specifically, we consider an environment of tasks provided with additional side information and we learn a conditioning function mapping task's side information into a task's specific meta-parameter vector. We then provide a statistical analysis demonstrating the potential advantage of our method over standard meta-learning.

**Contributions and organization.** Our work offers four contributions. First, in Sec. 2, we introduce a new conditional meta-learning framework with side information for biased regularization and fine tuning. Second, in Sec. 3, we formally show that, under certain assumptions, this conditional meta-learning approach results to be significantly advantageous w.r.t. the standard unconditional counterpart. We then describe two common settings in which such conditions are satisfied, supporting the potential importance of our study for real-world scenarios. Third, in Sec. 4, we propose a convex meta-algorithm providing a comparable advantage also in practice, as the number of observed tasks increases. Fourth, in Sec. 5, we present numerical experiments in which we test our theory and the performance of our method. Our conclusions are drawn in Sec. 6 and technical proofs are postponed to the appendix.

## 2   Conditional meta-learning

In this section we describe and contrast the conditional meta-learning setting with side information to standard meta-learning. We first introduce the class of inner algorithms we consider in this work.

**Inner algorithms (linear supervised learning).** Let $\mathcal{Z} = \mathcal{X} \times \mathcal{Y}$ with $\mathcal{X} \subseteq \mathbb{R}^d$ and $\mathcal{Y} \subseteq \mathbb{R}$ input and output spaces, respectively. Let $\mathcal{P}(\mathcal{Z})$ be the set of probability distributions (tasks) over $\mathcal{Z}$. Given $\mu \in \mathcal{P}(\mathcal{Z})$ and a loss function $\ell : \mathbb{R} \times \mathbb{R} \to \mathbb{R}$, our goal is to find a weight vector $w_\mu \in \mathbb{R}^d$ minimizing the *expected risk*

$$\min_{w \in \mathbb{R}^d} \mathcal{R}_\mu(w) \qquad \mathcal{R}_\mu(w) = \mathbb{E}_{(x,y) \sim \mu} \ell(\langle x, w \rangle, y), \tag{1}$$

where, $\langle x, w \rangle$ denotes the standard inner product between $x$ and $w \in \mathbb{R}^d$. In practice, $\mu$ is unknown and only accessible trough a training dataset $Z = (x_i, y_i)_{i=1}^n \sim \mu^n$ of i.i.d. (identically independently distributed) points sampled from $\mathcal{Z}$. The goal of a learning algorithm is to find a candidate weight vector incurring a small expected risk converging to the ideal $\mathcal{R}_\mu(w_\mu)$ as $n$ grows.

In this work we will focus on the family of learning algorithms performing biased regularized empirical risk minimization. Formally, given $\mathcal{D} = \bigcup_{n \in \mathbb{N}} \mathcal{Z}^n$ the space of all datasets (of any finite cardinality $n$) on $\mathcal{Z}$ and a bias vector $\theta \in \Theta = \mathbb{R}^d$, we will consider learning algorithms $A(\theta, \cdot) : \mathcal{D} \to \mathbb{R}^d$ such that,

$$A(\theta, Z) = \operatorname*{argmin}_{w \in \mathbb{R}^d} \mathcal{R}_{Z,\theta}(w) \qquad \mathcal{R}_{Z,\theta}(w) = \frac{1}{n} \sum_{i=1}^n \ell(\langle x_i, w \rangle, y_i) + \frac{\lambda}{2} \|w - \theta\|^2, \tag{2}$$

for any $Z = (x_i, y_i)_{i=1}^n$. Here $\|\cdot\|$ denotes the Euclidean norm on $\mathbb{R}^d$ and $\lambda > 0$ is a regularization parameter encouraging the algorithm $A(\theta, \cdot)$ to predict weight vectors that are close to $\theta$. We denote by $\mathcal{R}_Z(\cdot) = 1/n \sum_{i=1}^n \ell(\langle x_i, w \rangle, y_i)$ the empirical risk associated to $Z$.

**Remark 1** (Fine tuning). *In this work we primarily focus on the family (2) of batch inner algorithms. However, following [13, 14], it is possible to extend our analysis to fine-tuning algorithms performing online gradient descent on $\mathcal{R}_{Z,\theta}$, with starting point $w_1 = \theta \in \mathbb{R}^d$, namely*

$$A(\theta, Z) = \frac{1}{n} \sum_{i=1}^n w_i, \qquad w_{i+1} = w_i - \frac{s_i x_i - \lambda(w_i - \theta)}{\lambda i}, \qquad s_i \in \partial \ell(\cdot, y_i)(\langle x_i, w_i \rangle). \tag{3}$$

**(Unconditional) meta-learning.** Given a meta-distribution $\rho \in \mathcal{P}(\mathcal{M})$ (or *environment* [7]) over a family $\mathcal{M} \subseteq \mathcal{P}(\mathcal{Z})$ of distributions (tasks) $\mu$, meta-learning aims to learn an inner algorithm in the family that is well suited to tasks $\mu$ sampled from $\rho$. This goal can be reformulated as finding a meta-parameter $\theta_\rho \in \Theta$ whose associated algorithm $A(\theta_\rho, \cdot)$ minimizes the *transfer risk*

$$\min_{\theta \in \Theta} \mathcal{E}_\rho(\theta) \qquad \mathcal{E}_\rho(\theta) = \mathbb{E}_{\mu \sim \rho} \mathbb{E}_{Z \sim \mu^n} \mathcal{R}_\mu(A(\theta, Z)). \tag{4}$$

Standard meta-learning methods [5, 13, 14, 17–19, 24] usually address this problem via stochastic methods. They iteratively sample a task $\mu \sim \rho$ and a dataset $Z \sim \mu^n$, and, then, they perform a step of stochastic gradient descent on a surrogate problem of (4) computed by using $Z$.

Although remarkably effective in many applications [5, 13, 14, 17–19, 24], the framework above implicitly assumes that a single bias vector is sufficient for the entire family of tasks sampled from $\rho$. Since this assumption may not hold for more complex meta-distributions (e.g. multi-clusters), recent works have advocated a conditional perspective to tackle this problem [9, 22, 35, 39–41].

**Conditional meta-learning.** Assume now that when sampling a task $\mu$, we are also given additional side information $s \in \mathcal{S}$ to help solving the task. Within this setting the environment corresponds to a distribution $\rho \in \mathcal{P}(\mathcal{M}, \mathcal{S})$ over the set $\mathcal{M}$ of tasks and the set $\mathcal{S}$ of possible side information. The notion of side information is general, and recovers settings where $s$ contains descriptive features associated to a task (e.g. attributes in collaborative filtering [1]), additional information about the users in recommendation systems [20], or $s$ is an additional dataset sampled from $\mu$ (see [40] or Rem. 2 below). Intuitively, meta-learning might solve a new task better if it was able to leverage this additional side information. We formalize this concept by adapting (or conditioning) the meta-parameters $\theta \in \Theta$ on the side information $s \in \mathcal{S}$, by learning a meta-parameter-valued function $\tau$ minimizing

$$\min_{\tau \in \mathcal{T}} \mathcal{E}_\rho(\tau) \qquad \mathcal{E}_\rho(\tau) = \mathbb{E}_{(\mu,s)\sim\rho} \mathbb{E}_{Z\sim\mu^n} \mathcal{R}_\mu\big(A(\tau(s), Z)\big), \qquad (5)$$

over the space $\mathcal{T}$ of measurable functions $\tau : \mathcal{S} \to \Theta$. Note that the unconditional meta-learning problem in (4) is retrieved by restricting (5) to $\mathcal{T}^{\text{const}} = \{\tau \mid \tau(\cdot) \equiv \theta, \; \theta \in \Theta\}$, the set of constant functions associating any side information to a fixed bias vector. We assume $\rho$ to decompose in $\rho(\cdot|s)\rho_{\mathcal{S}}(\cdot)$ and $\rho(\cdot|\mu)\rho_{\mathcal{M}}(\cdot)$ the conditional and marginal distributions w.r.t. (with respect to) $\mathcal{S}$ and $\mathcal{M}$. In the following, we will quantify the benefits of adopting the conditional perspective above and, then, we propose an efficient algorithm to address (5). We conclude this section by drawing a connection between our formulation and previous work on the topic.

**Remark 2** (Datasets as side information). *A relevant setting is the case where the side information $s$ corresponds to an additional (conditional) dataset $Z^{cond}$ sampled from $\mu$, as proposed in [40]. We note however that our sampling scheme in (5) implies that side information $s$ and training set $Z$ are independent conditioned on $\mu$. Hence, our framework does not allow having $s = Z^{cond} = Z$, namely, to use the same dataset for both conditioning and training the inner algorithm $A(\tau(Z), Z)$, as done in [40]. This is a minor issue since one can always split $Z$ in two parts and use one part for training and the other one for conditioning.*

## 3    The advantage of conditional meta-learning

In this section we study the generalization properties of a given conditional function $\tau$. This will allow us to characterize the behavior of the ideal solution of (5) and to illustrate the potential advantage of conditional meta-learning. Specifically, we wish to estimate the error $\mathcal{E}_\rho(\tau)$ w.r.t. the ideal risk

$$\mathcal{E}_\rho^* = \mathbb{E}_{\mu\sim\rho} \mathcal{R}_\mu(w_\mu) \qquad w_\mu = \underset{w\in\mathbb{R}^d}{\operatorname{argmin}} \mathcal{R}_\mu(w). \qquad (6)$$

For any $\tau \in \mathcal{T}$ the following quantity will play a central role in our analysis:

$$\operatorname{Var}_\rho(\tau)^2 = \mathbb{E}_{(\mu,s)\sim\rho} \left\| w_\mu - \tau(s) \right\|^2. \qquad (7)$$

With some abuse of terminology, we refer to $\operatorname{Var}_\rho(\tau)$ as the *variance* of $w_\mu$ w.r.t. $\tau$ (it corresponds to the actual variance of $w_\mu$ when $\tau$ is the minimizer, see Lemma 2 below). Under the following assumption, we can control the excess risk of $\tau$ in terms of $\operatorname{Var}_\rho(\tau)$.

**Assumption 1.** *Let $\ell$ be a convex and $L$-Lipschitz loss function in the first argument. Additionally, there exist $R > 0$ such that $\|x\| \leq R$ for any $x \in \mathcal{X}$.*

**Theorem 1** (Excess risk with generic conditioning function $\tau$). *Let Asm. 1 hold. Given $\tau \in \mathcal{T}$, let $A(\theta, \cdot)$ be the generic inner algorithm in (2) with regularization parameter $\lambda = 2LR\operatorname{Var}_\rho(\tau)^{-1}n^{-1/2}$. Then,*

$$\mathcal{E}_\rho(\tau) - \mathcal{E}_\rho^* \leq \frac{2RL \operatorname{Var}_\rho(\tau)}{n^{1/2}}. \qquad (8)$$

**Proof.** We consider the decomposition $\mathcal{E}_\rho(\tau) - \mathcal{E}_\rho^* = \mathbb{E}_{(\mu,s)\sim\rho}[\mathrm{B}_{\mu,s} + \mathrm{C}_{\mu,s}]$, with

$$\mathrm{B}_{\mu,s} = \mathbb{E}_{Z\sim\mu^n}\left[\mathcal{R}_\mu(A(\tau(s),Z)) - \mathcal{R}_Z(A(\tau(s),Z))\right] \tag{9}$$

$$\mathrm{C}_{\mu,s} = \mathbb{E}_{Z\sim\mu^n}\left[\mathcal{R}_Z(A(\tau(s),Z)) - \mathcal{R}_\mu(w_\mu)\right] \leq \mathbb{E}_{Z\sim\mu^n}\left[\min_{w\in\mathbb{R}^d} \mathcal{R}_{Z,\tau(s)}(w) - \mathcal{R}_\mu(w_\mu)\right]. \tag{10}$$

$\mathrm{B}_{\mu,s}$ is the generalization error of the inner algorithm $A(\tau(s),\cdot)$ on the task $\mu$. Hence, applying Asm. 1 and the stability arguments in Prop. 5 in App. A, we can write $\mathrm{B}_{\mu,s} \leq 2R^2L^2(\lambda n)^{-1}$. Regarding the term $\mathrm{C}_{\mu,s}$, exploiting the definition of the algorithm in (2), we can write $\mathrm{C}_{\mu,s} \leq \frac{\lambda}{2}\|w_\mu - \tau(s)\|^2$. The desired statement follows by combining the two bounds above and optimizing w.r.t. $\lambda$. ∎

Thm. 1 suggests that a conditioning function $\tau$ with low variance can potentially incur a small excess risk. This makes the minimizer of the variance, a potentially good candidate for conditional meta-learning. We note that $\mathrm{Var}_\rho(\tau)$ in (6) can be interpreted as a Least-Squares risk associated to the input-(ideal) output pair $(s, w_\mu)$. Thanks to this interpretation, we can rely on the following well-known facts, see e.g. [11, Lemma A2].

**Lemma 2** (Best conditioning function in hindsight). *The minimizer of $\mathrm{Var}_\rho(\cdot)^2$ in (6) over the set $\mathcal{T}$ is such that $\tau_\rho(s) = \mathbb{E}_{\mu\sim\rho(\cdot|s)} w_\mu$ almost everywhere on $\mathcal{S}$. Moreover, for any $\tau \in \mathcal{T}$,*

$$\mathrm{Var}_\rho(\tau)^2 - \mathrm{Var}_\rho(\tau_\rho)^2 = \mathbb{E}_{s\sim\rho_{\mathcal{S}}} \left\|\tau(s) - \tau_\rho(s)\right\|^2. \tag{11}$$

Combining Thm. 1 with Lemma 2, we can formally analyze when the conditional approach is significantly advantageous w.r.t. the unconditional one.

**Conditional vs unconditional meta-learning.** As observed in (5), unconditional meta-learning consists in restricting to the class of constant conditioning functions $\mathcal{T}^{\mathrm{const}}$. Minimizing $\mathrm{Var}_\rho(\cdot)^2$ over this class yields the optimal bias vector for standard meta-learning (see e.g. [5, 13, 14, 24]), given by the expected target tasks' vector $w_\rho = \mathbb{E}_{\mu\sim\rho_{\mathcal{M}}} w_\mu$. Applying (11) to the constant function $\tau \equiv w_\rho$, we get the following gap between the best performance of conditional and unconditional meta-learning:

$$\mathrm{Var}_\rho(w_\rho)^2 - \mathrm{Var}_\rho(\tau_\rho)^2 = \mathbb{E}_{s\sim\rho_{\mathcal{S}}} \| w_\rho - \tau_\rho(s) \|^2. \tag{12}$$

We note that the gap (12) above is large when the ideal conditioning function $\tau_\rho$ is "far" from being the constant function $w_\rho$. We report below two examples that can be considered illustrative for many real-world scenarios in which such a condition is satisfied. We refer to App. B for the details and the deduction. In the examples, we parametrize each task with the triplet $\mu = (w_\mu, \eta_\mu, \xi_\mu)$, where $w_\mu$ is the target weight vector, $\eta_\mu$ is the marginal distribution on the inputs, $\xi_\mu$ is a noise model and $y \sim \mu(\cdot|x)$ is $y = \langle w_\mu, x \rangle + \epsilon$ with $x \sim \eta_\mu$ and $\epsilon \sim \xi_\mu$. Additionally, we denote by $\mathcal{N}(v, \sigma^2 I)$ a Gaussian distribution with mean $v \in \mathbb{R}^d$ and covariance matrix $\sigma^2 I$, with $I$ the $d \times d$ identity matrix.

**Example 1** (Clusters of tasks). *Let $\rho_{\mathcal{M}} = \frac{1}{m}\sum_{i=1}^m \rho_{\mathcal{M}}^{(m)}$ be a uniform mixture of $m$ environments (clusters) of tasks. For each $i = 1, \ldots, m$, a task $\mu \sim \rho_{\mathcal{M}}^{(i)}$ is sampled such that: 1) $w_\mu \sim \mathcal{N}(w(i), \sigma_w^2 I)$ with $w(i) \in \mathbb{R}^d$ a cluster's mean vector and $\sigma_w^2 I$ a covariance matrix, 2) the marginal $\eta_\mu = \mathcal{N}(x(i), \sigma_{\mathcal{X}}^2)$ with mean vector $x(i) \in \mathbb{R}^d$ and variance $\sigma_{\mathcal{X}}^2$, 3) the side information is an $n$ i.i.d. sample from $\eta_\mu$, namely $s = (x_i)_{i=1}^n \sim \eta_\mu^n$. Then, the gap between conditional and unconditional variance is*

$$\mathrm{Var}_\rho(w_\rho)^2 - \mathrm{Var}_\rho(\tau_\rho)^2 \geq \frac{1}{2m^2}\sum_{i,j=1}^m \left(1 - \frac{m}{2}e^{-\frac{n}{\sigma_{\mathcal{X}}^2}\|x(i)-x(j)\|^2}\right)\|w(i) - w(j)\|^2. \tag{13}$$

The inequality above confirms our natural intuition. It tells us that the larger is the number of clusters and the more the target weight vectors' and inputs' centroids are distant (i.e. the more the clusters are distant and the inputs' side information are discriminative for conditioning), the more the conditional approach will be advantageous w.r.t. the unconditional one.

**Example 2** (Curve of tasks). *Let $\rho_S$ be a uniform distribution over $S = [0,1]$. Let $h : S \to \mathbb{R}^d$ parametrize a circle of radius $r > 0$ centered in $c \in \mathbb{R}^d$, such as $h(s) = r\left(\cos(2\pi s), \sin(2\pi s), 0, \ldots, 0\right)^\top$. For $s \in S$, let $\mu \sim \rho(\cdot|s)$ such that $w_\mu \sim \mathcal{N}(h(s), \sigma^2 I)$ with $\sigma \in \mathbb{R}$. Then, $\tau_\rho = h$, $w_\rho = c$ and the the gap between conditional and unconditional variance is*

$$\mathrm{Var}_\rho(w_\rho)^2 - \mathrm{Var}_\rho(\tau_\rho)^2 = r^2. \tag{14}$$

Hence, in this case, the advantage in applying the conditional approach w.r.t. the unconditional one is equivalent to the squared radius of the circle over which the mean of the target weight vectors $w_\mu$ lie.

**Conditional meta-learning vs Independent Task Learning (ITL).** Solving each task independently corresponds to choosing the constant conditioning function $\tau_0 \equiv 0$. Applying Lemma 2 to this function, the gap between the performance of the best conditional approach and ITL reads as

$$\mathrm{Var}_\rho(0)^2 - \mathrm{Var}_\rho(\tau_\rho)^2 = \mathbb{E}_{s \sim \rho_S} \left\| w_\rho - \tau_\rho(s) \right\|^2 + \left\| w_\rho \right\|^2. \tag{15}$$

The gap in (15) combines the gain of conditional over unconditional meta-learning with $\|w_\rho\|^2 = \mathrm{Var}_\rho(0)^2 - \mathrm{Var}_\rho(w_\rho)^2$ that is the advantage of unconditional meta-learning over ITL (see [13, 14]). In the next section, we introduce a convex meta-algorithm mimicking this advantage also in practice.

# 4 Conditional meta-learning algorithm

To address conditional meta-learning in practice, we introduce the following set of conditioning functions. For a given feature map $\Phi : S \to \mathbb{R}^k$ on the side information space, we define the associated space of linear functions

$$\mathcal{T}_\Phi = \left\{ \tau : S \to \mathbb{R}^d \mid \tau(\cdot) = M\Phi(\cdot) + b, \text{ for some } M \in \mathbb{R}^{d \times k}, b \in \mathbb{R}^d \right\}. \tag{16}$$

To highlight the dependency of a function $\tau \in \mathcal{T}_\Phi$ w.r.t. its parameters $M$ and $b$, we will use the notation $\tau = \tau_{M,b}$. Evidently, $\mathcal{T}_\Phi$ contains the space of all unconditional estimators $\mathcal{T}^{\mathrm{const}}$. We consider $\mathcal{T}_\Phi$ equipped with the canonical norm $\|\tau_{M,b}\|^2 = \|(M,b)\|_F^2 = \|M\|_F^2 + \|b\|^2$, with $\|\cdot\|_F$ the Frobenius norm. We now introduce two standard assumptions will allow the design of our method.

**Assumption 2.** *The minimizer $\tau_\rho$ of $\mathrm{Var}_\rho(\cdot)$ belongs to $\mathcal{T}_\Phi$, namely there exist $M_\rho \in \mathbb{R}^{d \times k}$ and $b_\rho \in \mathbb{R}^d$, such that $\tau_\rho(\cdot) = M_\rho\Phi(\cdot) + b_\rho$.*

**Assumption 3.** *There exists $K > 0$ such that $\|\Phi(s)\| \le K$ for any $s \in S$.*

Asm. 2 enables us to restrict the conditional meta-learning problem in (5) to $\mathcal{T}_\Phi$, rather than to the entire space $\mathcal{T}$ of measurable functions. In Lemma 7 in App. C we provide the closed forms of $M_\rho$ and $b_\rho$ and we express the gap in (12) by the correlation between $w_\mu$ and $\Phi(s)$ and the slope of $\tau_\rho$. Asm. 3 will allow us to work with a Lipschitz meta-objective, as explained below.

**The convex surrogate problem.** Following a similar strategy to the one adopted for the unconditional setting in [13, 14], we introduce the following surrogate problem for the conditional one in (5):

$$\min_{\tau \in \mathcal{T}} \hat{\mathcal{E}}_\rho(\tau) \qquad \hat{\mathcal{E}}_\rho(\tau) = \mathbb{E}_{(\mu,s) \sim \rho} \mathbb{E}_{Z \sim \mu^n} \mathcal{R}_{Z,\tau(s)}(A(\tau(s), Z)), \tag{17}$$

where we have replaced the inner expected risk $\mathcal{R}_\mu$ with the regularized empirical risk $\mathcal{R}_{Z,\theta}$ in (2). Exploiting Asm. 2, the problem above can be rewritten more explicitly as follows

$$\min_{M \in \mathbb{R}^{d \times k}, b \in \mathbb{R}^d} \mathbb{E}_{(\mu,s) \sim \rho} \mathbb{E}_{Z \sim \mu^n} \mathcal{L}\big(M, b, s, Z\big) \quad \mathcal{L}\big(M, b, s, Z\big) = \mathcal{R}_{Z,\tau_{M,b}(s)}(A(\tau_{M,b}(s), Z)). \tag{18}$$

The following proposition characterizes useful properties of the meta-loss $\mathcal{L}(\cdot, \cdot, s, Z)$ introduced above (such as convexity and differentiability) and it supports its choice as surrogate meta-loss. We denote by $\cdot^\top$ the standard transposition operation.

**Proposition 3** (Properties of the surrogate meta-loss $\mathcal{L}$). *For any $Z \in \mathcal{D}$ and $s \in S$, the function $\mathcal{L}(\cdot, \cdot, s, Z)$ is convex, differentiable and its gradient is given by*

$$\nabla \mathcal{L}\big(\cdot, \cdot, s, Z\big)(M, b) = -\lambda \Big( A\big(\tau_{M,b}(s), Z\big) - \tau_{M,b}(s) \Big) \begin{pmatrix} \Phi\big(s\big) \\ 1 \end{pmatrix}^\top \tag{19}$$

*for any $M \in \mathbb{R}^{d \times k}$ and $b \in \mathbb{R}^d$. Moreover, under Asm. 1 and Asm. 3, we have*

$$\left\| \nabla \mathcal{L}\big(\cdot, \cdot, s, Z\big)(M, b) \right\|_F^2 \le L^2 R^2 (K^2 + 1). \tag{20}$$

---

**Algorithm 1** Meta-Algorithm, SGD on (18)

---

**Input** $\gamma > 0$ meta-step size, $\lambda > 0$ inner regularization parameter
**Initialization** $M_1 = 0 \in \mathbb{R}^{d \times k}$, $b_1 = 0 \in \mathbb{R}^d$
**For** $t = 1$ to $T$
     Receive $(\mu_t, s_t) \sim \rho$ and $Z_t \sim \mu_t^n$
     Let $\theta_t = \tau_{M_t, b_t}(s_t) = M_t \Phi(s_t) + b_t$
     Run the inner algorithm in (2) to obtain $w_t = A(\theta_t, Z_t)$
     Compute $\nabla \mathcal{L}(\cdot, \cdot, s_t, Z_t)(M_t, b_t) = -\lambda(w_t - \theta_t) \left( \begin{array}{c} \Phi(s_t) \\ 1 \end{array} \right)^{\top}$ as in (19)
     Update $(M_{t+1}, b_{t+1}) = (M_t, b_t) - \gamma \nabla \mathcal{L}(\cdot, \cdot, s_t, Z_t)(M_t, b_t)$
**Return** $\bar{M} = \frac{1}{T} \sum_{t=1}^{T} M_t$, $\bar{b} = \frac{1}{T} \sum_{t=1}^{T} b_t$

---

The proof of Prop. 3 is reported in App. D.1 and it follows a similar reasoning in [14], by taking into account also the parameter $M$ in the optimization problem.

**The conditional meta-learning estimator.** In this work we propose to apply Stochastic Gradient Descent (SGD) on the surrogate problem in (18). Alg. 1 summarizes the implementation of this approach: assuming a sequence of i.i.d. pairs $(Z_t, s_t)_{t=1}^T$ of training sets and side information, at each iteration the algorithm updates the conditional iterates $(M_t, b_t)$ by performing a step of constant size $\gamma > 0$ in the direction of $-\nabla \mathcal{L}(\cdot, \cdot, s_t, Z_t)(M_t, b_t)$. The map $\tau_{\bar{M}, \bar{b}}$ is then returned as conditional estimator, with $(\bar{M}, \bar{b})$ the average across all the iterates $(M_t, b_t)_{t=1}^T$. The following result characterizes the excess risk of the proposed estimator.

**Theorem 4** (Excess risk bound for the conditioning function returned by Alg. 1). *Let Asm. 1 and Asm. 3 hold. Let $\tau_{M,b}$ be a fixed function in $\mathcal{T}_\Phi$ and let $\mathrm{Var}_\rho(\tau_{M,b})^2$ be the corresponding variance introduced in (7). Let $\bar{M}$ and $\bar{b}$ be the outputs of Alg. 1 applied to a sequence $(Z_t, s_t)_{t=1}^T$ of i.i.d. pairs sampled from $\rho$ with inner regularization parameter and meta-step size*

$$\lambda = \frac{2RL}{\mathrm{Var}_\rho(\tau_{M,b})} \frac{1}{\sqrt{n}} \qquad \gamma = \frac{\|(M,b)\|_F}{LR\sqrt{(K^2+1)}} \frac{1}{\sqrt{T}}. \tag{21}$$

*Then, in expectation w.r.t. the sampling of $(Z_t, s_t)_{t=1}^T$,*

$$\mathbb{E} \, \mathcal{E}_\rho(\tau_{\bar{M}, \bar{b}}) - \mathcal{E}_\rho^* \leq \frac{2RL \mathrm{Var}_\rho(\tau_{M,b})}{\sqrt{n}} + \frac{LR\sqrt{K^2+1} \, \|(M,b)\|_F}{\sqrt{T}}. \tag{22}$$

**Proof (Sketch).** We consider the following decomposition

$$\mathbb{E} \, \mathcal{E}_\rho(\tau_{\bar{M}, \bar{b}}) - \mathcal{E}_\rho^* = \underbrace{\mathbb{E} \, \mathcal{E}_\rho(\tau_{\bar{M}, \bar{b}}) - \hat{\mathcal{E}}_\rho(\tau_{\bar{M}, \bar{b}})}_{\text{B}} + \underbrace{\mathbb{E} \, \hat{\mathcal{E}}_\rho(\tau_{\bar{M}, \bar{b}}) - \hat{\mathcal{E}}_\rho(\tau_{M,b})}_{\text{C}} + \underbrace{\hat{\mathcal{E}}_\rho(\tau_{M,b}) - \mathcal{E}_\rho^*}_{\text{D}}. \tag{23}$$

Applying Asm. 1 and the stability arguments in Prop. 5 in App. A, we can write B $\leq 2R^2L^2(\lambda n)^{-1}$. The term C is the term expressing the convergence rate of Alg. 1 on the surrogate problem in (18) and, exploiting Asm. 3 and Prop. 3, it can be controlled as described in Prop. 9 in App. D.2. Regarding the term D, exploiting the definition of the algorithm in (2), we can write D $\leq \frac{\lambda}{2} \mathrm{Var}_\rho(\tau_{M,b})^2$. Combining all the terms and optimizing w.r.t. $\gamma$ and $\lambda$, we get the desired statement. ∎

We now comment about the result we got above in Thm. 4.

**Proposed vs optimal conditioning function.** Specializing the bound in Thm. 4 to the best conditioning function $\tau_\rho$ in Lemma 2, thanks to Asm. 2, we get the following bound for our estimator:

$$\mathbb{E} \, \mathcal{E}_\rho(\tau_{\bar{M}, \bar{b}}) - \mathcal{E}_\rho^* \leq \mathcal{O}\Big( \mathrm{Var}_\rho(\tau_\rho) \, n^{-1/2} + \|(M_\rho, b_\rho)\|_F \, T^{-1/2} \Big). \tag{24}$$

Hence, our proposed meta-algorithm achieves comparable performance to the best conditioning function $\tau_\rho$ in hindsight, provided that the number of observed tasks is sufficiently large. The bound

above also highlights the trade-off between statistical and computational complexity of the class $\mathcal{T}_\phi$: conditional meta-learning incurs in a cost $\|(M_\rho, b_\rho)\|_F$ in the $\sqrt{T}$-term that is larger than the $\|b_\rho\|$ cost of unconditional meta-learning (see [5, 13, 24]), which is, however, limited to constant conditioning functions. This is an acceptable price, since, as we discussed in Sec. 3, the performance of conditional meta-learning is significantly better than the standard one in many common scenarios.

**Remark 3.** *When $\tau_\rho \notin \mathcal{T}_\Phi$ (i.e. when Asm. 3 does not hold), our method suffers an additional approximation error due to the fact $\min_{\tau \in \mathcal{T}_\Phi} \mathrm{Var}_\rho(\tau) > \mathrm{Var}_\rho(\tau_\rho)$. In this case, one might nullify the gap above by considering a feature map $\Phi : \mathcal{S} \to \mathcal{H}$ with $\mathcal{H}$ a universal reproducing kernel Hilbert space of functions. Exploiting standard arguments from online learning with kernels literature (see e.g. [25, 36, 37]), in Lemma 10 in App. D.3 we describe the implementation of Alg. 1 for this setting using only evaluations of the kernel associated to the feature map. We leave the corresponding theoretical analysis to future work.*

**Proposed conditioning function vs unconditional meta-learning.** Specializing Thm. 4 to $\tau_{M,b} \equiv w_\rho$, the bound for our estimator becomes:

$$\mathbb{E}\,\mathcal{E}_\rho(\tau_{\overline{M},\overline{b}}) - \mathcal{E}_\rho^* \leq \mathcal{O}\big(\mathrm{Var}_\rho(w_\rho)\,n^{-1/2} + \|w_\rho\|\,T^{-1/2}\big), \tag{25}$$

which is equivalent to state-of-the-art bounds for unconditional methods, see [5, 13, 14, 24]. Hence, our conditional approach provides, at least, the same guarantees as its unconditional counterpart.

**Proposed conditioning function vs ITL.** Specializing Thm. 4 to $\tau_{M,b} \equiv 0$ corresponds to force $\gamma = 0$ and, consequently, Alg. 1 to not move. In such a case, we get the bound:

$$\mathbb{E}\,\mathcal{E}_\rho(\tau_{\overline{M},\overline{b}}) - \mathcal{E}_\rho^* \leq \mathcal{O}\big(\mathrm{Var}_\rho(0)\,n^{-1/2}\big), \tag{26}$$

which corresponds to the standard excess risk bound for ITL, see [5, 13, 14, 24]. In other words, our method does not generate negative transfer effect.

**Remark 4** (Fine tuning). *In the case of the online inner family in Rem. 1 used in fine tuning, Alg. 1 employs an approximation of the meta-subgradient in (19) by replacing the batch regularized empirical risk minimizer $A(\tau_{M,b}(s), Z)$ in (2) with the last iterate of the online algorithm in (3). As shown in [13, 14] for the unconditional setting, such an approximation does not affect the behavior of the bounds above.*

## 5 Experiments

In this section we compare the numerical performance [1] of our conditional method in Alg. 1 (cond.) w.r.t. its unconditional counterpart in [13] (uncond.). We will also add to the comparison the methods consisting in applying the inner algorithm on each task with $\tau \equiv 0 \in \mathbb{R}^d$ (i.e. ITL) and the unconditional oracle $\tau \equiv w_\rho = \mathbb{E}_{\mu \sim \rho_{\mathcal{M}}}\,w_\mu$ (mean), when available. We considered regression problems and we evaluated the errors by the absolute loss. The results refer to the fine tuning variant of the methods with the online inner algorithm in (3). In App. E, we describe how we tuned the hyper-parameters $\lambda, \gamma$ in our experiments.

**Synthetic clusters.** We considered three variants of the setting described in Ex. 1. In all the variants we sampled $T_{\mathrm{tot}} = 480$ tasks from a mixture of $m$ clusters with the same probability. For each task $\mu$, we sampled the corresponding target vector $w_\mu$ from the $d = 20$-dimensional Gaussian distribution $\mathcal{N}(w(j_\mu), I)$, where, $j_\mu \in \{1, \ldots, m\}$ denotes the cluster from which the task $\mu$ was sampled. We then generated the corresponding dataset $(x_i, y_i)_{i=1}^{n_{\mathrm{tot}}}$ with $n_{\mathrm{tot}} = 20$. We sampled the inputs from $\mathcal{N}(x(j_\mu), I)$ and we generated the labels according to the equation $y = \langle x, w_\mu \rangle + \epsilon$, with the noise $\epsilon$ sampled from $\mathcal{N}(0, \sigma^2 I)$, with $\sigma$ chosen in order to have signal-to-noise ratio equal to 1.
We implemented our conditional method using as side information the training input points $X = (x_i)_{i=1}^{n_{\mathrm{tr}}} \in \bigcup_{n \in \mathbb{N}} \mathcal{X}^n$ and the feature map $\Phi : \bigcup_{n \in \mathbb{N}} \mathcal{X}^n \to \mathbb{R}^d$ defined by $\Phi(X) = \frac{1}{n_{\mathrm{tr}}} \sum_{i=1}^{n_{\mathrm{tr}}} x_i$.
In Fig. 1 (left-top), we generated an environment as above with just one cluster ($m = 1$) and we took $w(1) = 4 \in \mathbb{R}^d$ (the vector in $\mathbb{R}^d$ with all components 4) and $x(1) = 1 \in \mathbb{R}^d$. As we can see, coherently with previous work [13], the unconditional approach outperforms ITL and it converges to the mean vector $w_\rho = w(1)$ as the number of training tasks increases. The conditional approach returns equivalent performances to the unconditional counterpart.

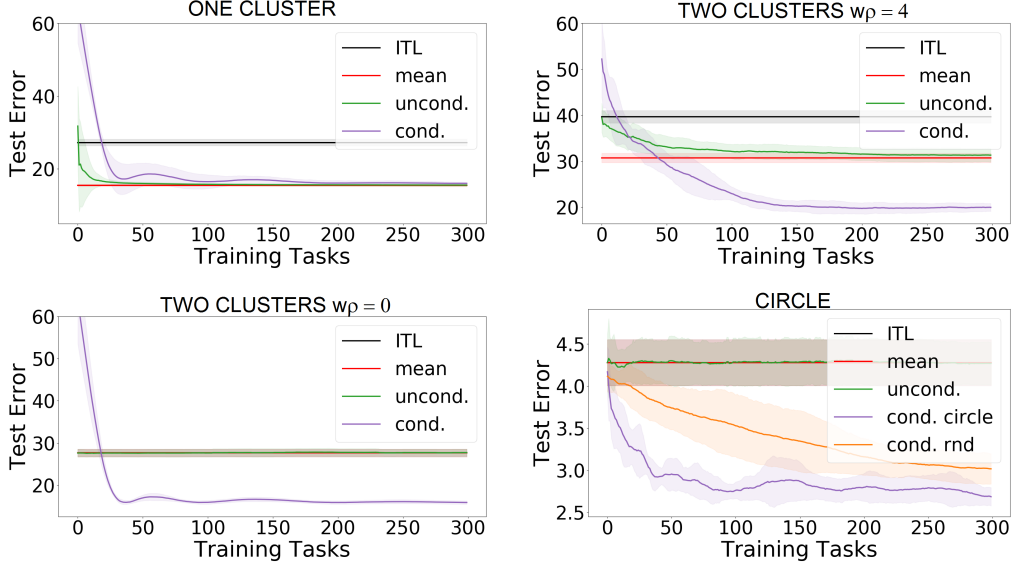

Figure 1: Performance (averaged over 10 seeds) of different methods w.r.t. an increasing number of tasks on different environments: with one cluster (left-top), with two clusters and $w_\rho = 4$ (right-top), with two clusters and $w_\rho = 0$ (left-bottom), circle (right-bottom).

In Fig. 1 (right-top), we considered an environment of two clusters ($m = 2$) identified by $w(1) = 8 \in \mathbb{R}^d$, $w(2) = 0 \in \mathbb{R}^d$ (implying $w_\rho = 4$), $x(1) = 1 \in \mathbb{R}^d$ and $x(2) = -x(1)$. As we can see, the conditional approach outperform ITL as in the previous setting, but the conditional approach yields even better performance.

Finally, in Fig. 1 (left-bottom), we considered an environment of two clusters ($m = 2$) identified by $w(1) = 4 \in \mathbb{R}^d$, $w(2) = -w(1)$ (implying $w_\rho = 0$), $x(1) = 1 \in \mathbb{R}^d$ and $x(2) = -x(1)$. As expected, the unconditional approach mimics the poor performance of ITL, while, the performance of the conditional approach is promising.

Summarizing, the conditional approach brings advantage w.r.t. the unconditional one when the heterogeneity of the environment is significant. When the environment is homogeneous, the performance of the two are equivalent. This conclusion is exactly inline with our theory in (25) and (26).

**Synthetic circle.** We sampled $T_{\text{tot}} = 480$ tasks according to the setting described in Ex. 2. Specifically, for each task $\mu$, we first sampled the corresponding side information $s \in [0, 1]$ according to the uniform distribution. We then generated the vector $h(s) = r \, (\cos(2\pi s), \sin(2\pi s), 0, \ldots, 0)^\top \in \mathbb{R}^d$, with $d = 20$, on the zero-centered circle of radius $r = 8$. After this, we sampled the corresponding target weight vector $w_\mu$ from $\mathcal{N}(h(s), I)$. We then generated the associated dataset of $n_{\text{tot}} = 20$ points as for the experiments above. We applied our conditional approach with the true underlying feature map $\Phi(s) = (\cos(2\pi s), \sin(2\pi s))$ (cond. circle) and the feature map mimicking a Gaussian distribution by Fourier random features [34] described below (at the end of this section) with parameters $k = 50$ and $\sigma = 10$ (cond. rnd).

From Fig. 1 (right-bottom) we see that the performance of unconditional meta-learning mimics the poor performance of ITL (in fact, we have $w_\rho = 0$). On the other hand, both the conditional approaches bring a substantial advantage and the random features' variant approaches the variant knowing the true underlying feature map.

**Lenk dataset.** We considered the computer survey data from [26, 30], in which $T_{\text{tot}} = 180$ people (tasks) rated the likelihood of purchasing one of $n_{\text{tot}} = 20$ different personal computers. The input represents $d = 13$ different computers' characteristics, while the output is an integer rating from 0 to 10. In this case, we used as side information the training datapoints $Z = (z_i)_{i=1}^{n_{\text{tr}}}$ and the feature map $\Phi : \mathcal{D} \to \mathbb{R}^{2d}$ defined by $\Phi(Z) = \frac{1}{n_{\text{tr}}} \sum_{i=1}^{n_{\text{tr}}} \phi(z_i)$, with $\phi(z_i) = \text{vec}(x_i(y_i, 1)^\top)$, where, for any matrix $A = [a_1, a_2] \in \mathbb{R}^{d \times 2}$ with columns $a_1, a_2 \in \mathbb{R}^d$, $\text{vec}(A) = (a_1, a_2)^\top \in \mathbb{R}^{2d}$. Fig. 2 (left) shows that, coherently to previous literature [13], the unconditional approach significantly outperforms ITL, but the performance of its conditional counterpart is even better.

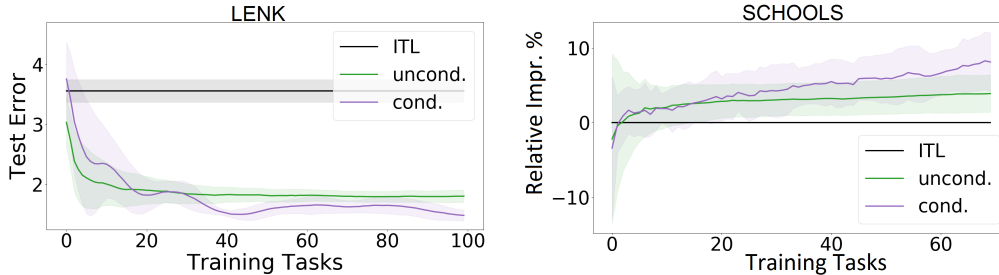

Figure 2: Performance (averaged over 10 seeds) of different methods w.r.t. an increasing number of tasks. Lenk dataset (left), Schools dataset (right).

**Schools dataset.** We considered the Schools dataset [3], consisting of examination records from $T_{\text{tot}} = 139$ schools. Each school is associated to a task, individual students are represented by a features' vectors $x \in \mathbb{R}^d$, with $d = 26$, and their exam scores to the outputs. The sample size $n_{\text{tot}}$ varies across the tasks from a minimum 24 to a maximum 251. We used as side information the training inputs $X = (x_i)_{i=1}^{n_{\text{tr}}}$ and the feature map mimicking a Gaussian distribution by Fourier random features described below (at the end of this section) with parameters $k = 1000$ and $\sigma = 100$. Fig. 2 (right) shows that, also in this case, the unconditional approach brings a meaningful improvement w.r.t. ITL, but the gain provided by its conditional counterpart is even more evident.

**Feature map by Fourier random features.** We now describe the feature map mimicking a Gaussian distribution by Fourier random features [34] we used in our synthetic circle experiment and Schools dataset experiment. We recall that, in these cases, we considered as side information the inputs $X = (x_i)_{i=1}^{n}$. The feature map above was then defined as $\Phi(X) = \frac{1}{n} \sum_{i=1}^{n} \phi(x_i)$, where, $\phi$ was built as follows. We first introduced an integer $k \in \mathbb{N}$ and a constant $\sigma \in \mathbb{R}$. We then sampled a vector $v \in \mathbb{R}^k$ from the uniform distribution over $[0, 2\pi]^k$ and a matrix $U \in \mathbb{R}^{k \times d}$ is sampled from the Gaussian distribution $\mathcal{N}(0, \sigma I)$. We then defined

$$\phi(x_i) = \sqrt{\frac{2}{k}} \cos(Ux_i + v) \in \mathbb{R}^k, \tag{27}$$

where $\cos(\cdot)$ is applied component-wise to the vector.

## 6 Conclusion

We proposed a new conditional meta-learning framework for biased regularization and fine tuning based on side information and we provided a theoretical analysis demonstrating its potential advantage over standard meta-learning, when the environment of tasks is heterogeneous. In the future, taking inspiration from [12, 32], it would be interesting to develop a variant of our method in which the hyper-parameters are automatically tuned in efficient way. In addition, it would valuable to extend our conditional approach and the corresponding analysis to other meta-learning paradigms considering different families of inner algorithms, such as [14, 38].

## Broader impact

Meta-learning is a very important field for machine learning with potential societal implications related to applications such as recommender systems. In this work we focused mostly on theoretical and modeling aspects, however in the future the topic will need to take into consideration contributions from other fields related to ethical and societal aspects, such as privacy and fairness.

## Acknowledgments and Disclosure of Funding

This work was supported in part by SAP SE and EPSRC Grant N. EP/P009069/1. C.C. acknowledges the Royal Society (grant SPREM RGS\R1\201149).

## Footnotes

[1]Code is available at https://github.com/dGiulia/ConditionalMetaLearning.git

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
