[Supplementary Material · camera_ready_conditional_metalearning.pdf]

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

# Appendix

The supplementary material is organized as follows. In App. A we give the bound on the generalization error of the algorithm in (2) that we used in various proofs. In App. B we formally describe the deduction of the statements reported in Ex. 1 and Ex. 2 in Sec. 3. In App. C we report the closed form of $M_\rho \in \mathbb{R}^{d \times k}$ and $b_\rho \in \mathbb{R}^d$ in Asm. 2 and we express the gap between the conditional and the unconditional variance in (12) by the correlation between the target tasks' vectors $w_\mu$ and the transformed side information $\Phi(s)$ or the slope of $\tau_\rho$. In App. D, we report the proofs of the statements we used in Sec. 4 in order to prove the expected excess risk bound in Thm. 4 for Alg. 1. Finally, in App. E, we report the experimental details we omitted in the main body.

## A Generalization error of the algorithm in (2)

In this section we report the generalization error bound of the family of inner algorithms in (2) that we used in our proofs. The statement exploits standard tools from stability theory. We do not claim any originality, we report the proof for completeness.

**Proposition 5** (Generalization error of the algorithm in (2)). *For a distribution $\mu \sim \rho$, fix a dataset $Z = (x_i, y_i)_{i=1}^n \sim \mu^n$ and, for any $i \in \{1, \dots, n\}$, fix a datapoint $z_i' = (x_i', y_i') \sim \mu$ independent from $Z$. For any $\theta \in \Theta$ not depending on $Z$, let $\hat{w}_\theta(Z) = A(\theta, Z)$ be the output of the algorithm in (2) over $Z$ and let $s_{\theta,i}' \in \partial \ell(\cdot, y_i')(\langle x_i', \hat{w}_\theta(Z)\rangle)$ be a subgradient of $\ell(\cdot, y_i')$ at $\langle x_i', \hat{w}_\theta(Z)\rangle$. Then, the following generalization error bound holds for $\hat{w}_\theta(Z)$*

$$\mathbb{E}_{Z \sim \mu^n} \left[ \mathcal{R}_\mu(\hat{w}_\theta(Z)) - \mathcal{R}_Z(\hat{w}_\theta(Z)) \right] \leq \frac{2}{\lambda n} \mathbb{E}_{Z \sim \mu^n} \mathbb{E}_{z_i' \sim \mu} \left\| x_i' s_{\theta,i}' \right\|^2. \tag{28}$$

*As a consequence, under Asm. 1, the right side term above can be upper bounded by $2L^2 R^2 (\lambda n)^{-1}$.*

**Proof.** For any $i \in \{1, \dots, n\}$, consider the dataset $Z^{(i)}$, a copy of the original dataset $Z$ in which we exchange the point $z_i = (x_i, y_i)$ with the new i.i.d. point $z_i' = (x_i', y_i')$. For a fixed $\theta \in \Theta$, we analyze how much this perturbation affects the outputs of the algorithm in (2). In other words, we study the discrepancy between $\hat{w}_\theta(Z)$ and $\hat{w}_\theta(Z^{(i)})$. We start from observing that, since $\mathcal{R}_{Z,\theta}$ is $\lambda$-strongly convex w.r.t. $\| \cdot \|$, by growth condition and the definition of the algorithm in (2), we can write the following

$$\begin{aligned} \frac{\lambda}{2} \left\| \hat{w}_\theta(Z^{(i)}) - \hat{w}_\theta(Z) \right\|^2 &\leq \mathcal{R}_{Z,\theta}(\hat{w}_\theta(Z^{(i)})) - \mathcal{R}_{Z,\theta}(\hat{w}_\theta(Z)) \\ \frac{\lambda}{2} \left\| \hat{w}_\theta(Z^{(i)}) - \hat{w}_\theta(Z) \right\|^2 &\leq \mathcal{R}_{Z^{(i)},\theta}(\hat{w}_\theta(Z)) - \mathcal{R}_{Z^{(i)},\theta}(\hat{w}_\theta(Z^{(i)})). \end{aligned} \tag{29}$$

Hence, summing the two inequalities above, we get

$$\begin{aligned} \lambda \left\| \hat{w}_\theta(Z^{(i)}) - \hat{w}_\theta(Z) \right\|^2 &\leq \mathcal{R}_{Z,\theta}(\hat{w}_\theta(Z^{(i)})) - \mathcal{R}_{Z^{(i)},\theta}(\hat{w}_\theta(Z^{(i)})) + \mathcal{R}_{Z^{(i)},\theta}(\hat{w}_\theta(Z)) - \mathcal{R}_{Z,\theta}(\hat{w}_\theta(Z)) \\ &= \frac{\mathbf{B} + \mathbf{C}}{n}, \end{aligned} \tag{30}$$

where we have introduced the terms

$$\begin{aligned} \mathbf{B} &= \ell(\langle x_i', \hat{w}_\theta(Z)\rangle, y_i') - \ell(\langle x_i', \hat{w}_\theta(Z^{(i)})\rangle, y_i') \\ \mathbf{C} &= \ell(\langle x_i, \hat{w}_\theta(Z^{(i)})\rangle, y_i) - \ell(\langle x_i, \hat{w}_\theta(Z)\rangle, y_i). \end{aligned} \tag{31}$$

Now, exploiting the assumption $s_{\theta,i}' \in \partial \ell(\cdot, y_i')(\langle x_i', \hat{w}_\theta(Z)\rangle)$, applying Holder's inequality and introducing a subgradient $s_{\theta,i} \in \partial \ell(\cdot, y_i)(\langle x_i, \hat{w}_\theta(Z^{(i)})\rangle)$, we can write

$$\begin{aligned} \mathbf{B} &\leq \langle x_i' s_{\theta,i}', \hat{w}_\theta(Z) - \hat{w}_\theta(Z^{(i)})\rangle \leq \left\| x_i' s_{\theta,i}' \right\| \left\| \hat{w}_\theta(Z^{(i)}) - \hat{w}_\theta(Z) \right\| \\ \mathbf{C} &\leq \langle x_i s_{\theta,i}, \hat{w}_\theta(Z^{(i)}) - \hat{w}_\theta(Z)\rangle \leq \left\| x_i s_{\theta,i} \right\| \left\| \hat{w}_\theta(Z^{(i)}) - \hat{w}_\theta(Z) \right\|. \end{aligned} \tag{32}$$

Combining these last two inequalities with (30) and simplifying, we get the following

$$\left\| \hat{w}_\theta(Z^{(i)}) - \hat{w}_\theta(Z) \right\| \leq \frac{1}{\lambda n} \left( \left\| x_i' s_{\theta,i}' \right\| + \left\| x_i s_{\theta,i} \right\| \right). \tag{33}$$

Hence, combining the first row in (32) with (33), we can write

$$\ell(\langle x'_i, \hat{w}_\theta(Z)\rangle, y'_i) - \ell(\langle x'_i, \hat{w}_\theta(Z^{(i)})\rangle, y'_i) \le \frac{1}{\lambda n}\left(\left\|x'_i s'_{\theta,i}\right\|^2 + \left\|x'_i s'_{\theta,i}\right\|\left\|x_i s_{\theta,i}\right\|\right). \tag{34}$$

Now, taking the expectation w.r.t. $Z \sim \mu^n$ and $z'_i \sim \mu$ of the left side member above, according to [8, Lemma 7], we get

$$\mathbb{E}_{Z\sim\mu^n}\mathbb{E}_{z'_i\sim\mu}\left[\ell(\langle x'_i, \hat{w}_\theta(Z)\rangle, y'_i) - \ell(\langle x'_i, \hat{w}_\theta(Z^{(i)})\rangle, y'_i)\right] = \mathbb{E}_{Z\sim\mu^n}\left[\mathcal{R}_\mu(\hat{w}_\theta(Z)) - \mathcal{R}_Z(\hat{w}_\theta(Z))\right].$$

Finally, taking the expectation of the right side member, exploiting the fact that the points are i.i.d. according $\mu$, we get

$$\mathbb{E}_{Z\sim\mu^n}\mathbb{E}_{z'_i\sim\mu}\frac{1}{\lambda n}\left(\left\|x'_i s'_{\theta,i}\right\|^2 + \left\|x'_i s'_{\theta,i}\right\|\left\|x_i s_{\theta,i}\right\|\right) \le \frac{2}{\lambda n}\mathbb{E}_{Z\sim\mu^n}\mathbb{E}_{z'_i\sim\mu}\left\|x'_i s'_{\theta,i}\right\|^2, \tag{35}$$

where we recall that $s'_{\theta,i} \in \partial\ell(\cdot, y'_i)(\langle x'_i, \hat{w}_\theta(Z)\rangle)$. The statement derives from combining the two last statements above with the expectation w.r.t. $Z \sim \mu^n$ and $z'_i \sim \mu$ of (34). The second statement directly derives from the first one, once one observes that, if $\ell(\cdot, y)$ is $L$-Lipschitz for any $y \in \mathcal{Y}$, then, $|s'_{\theta,i}| \le L$ (see [36, Lemma 14.7]). ∎

## B  Examples

In this section, we provide the deduction of the statements in the examples reported in Sec. 3. We start from presenting some computation regarding a generic environment parametrized by a latent variable in App. B.1 and, then, in App. B.2, we specify this computation and we derive the statement in Ex. 1. Finally, in App. B.3, we prove the statement in Ex. 2.

### B.1  General parametrization

Consider the case where a latent variable $\alpha \in \mathcal{A}$ parametrizes the environment $\rho$. Denote by $\rho(\cdot|\alpha)$ the conditional distributions given $\alpha$ and by $\rho_{\mathcal{A}}$ the marginal distribution of the latent variable. As usual, we assume $\rho(\mu, \alpha) = \rho(\mu|\alpha)\rho_{\mathcal{A}}(\alpha)$. Introduce also

$$w(\alpha) = \int w_\mu \, \rho(\mu|\alpha) \, dw_\mu \qquad \sigma(\alpha)^2 = \frac{1}{2}\int \|w_\mu - w(\alpha)\|^2 \, \rho(\mu|\alpha) \, dw_\mu \tag{36}$$

the conditional expectation and the conditional variance of the target weight vectors $w_\mu$ given $\alpha$, respectively. We now explicitly compute the unconditional and the conditional variance for this generic environment.

**Unconditional variance.** We start from observing that thanks to the parametrization of the environment $\rho$, we can rewrite the unconditional variance as follows

$$\text{Var}_\rho(w_\rho)^2 = \mathbb{E}_{\mu\sim\rho_{\mathcal{M}}}\|w_\mu - w_\rho\|^2 = \int \|w_\mu - w_\rho\|^2 \, \rho(\mu) \, dw_\mu$$

$$= \int\left(\int \|w_\mu - w_\rho\|^2 \, \rho(\mu|\alpha) \, dw_\mu\right)\rho_{\mathcal{A}}(\alpha) \, d\alpha. \tag{37}$$

We now observe that, for any $\alpha \in \mathcal{A}$, we can write the following

$$\int \|w_\mu - w_\rho\|^2 \, \rho(\mu|\alpha) \, dw_\mu$$

$$= \int\left(\|w_\mu\|^2 - 2\langle w_\mu, w_\rho\rangle + \|w_\rho\|^2\right)\rho(\mu|\alpha) \, dw_\mu$$

$$= \int \|w_\mu\|^2 \, \rho(\mu|\alpha) \, dw_\mu - 2\langle w(\alpha), w_\rho\rangle + \|w_\rho\|^2$$

$$= \int \|w_\mu\|^2 \, \rho(\mu|\alpha) \, dw_\mu \pm \|w(\alpha)\|^2 - 2\langle w(\alpha), w_\rho\rangle + \|w_\rho\|^2 \tag{38}$$

$$= \int \|w_\mu - w(\alpha)\|^2 \, \rho(\mu|\alpha) \, dw_\mu + \|w(\alpha) - w_\rho\|^2$$

$$= 2\sigma(\alpha)^2 + \|w(\alpha) - w_\rho\|^2.$$

Hence, substituting in (37), we get

$$\mathrm{Var}_\rho(w_\rho)^2 = 2\int \sigma(\alpha)^2 \, \rho_\mathcal{A}(\alpha) \, d\alpha + \int \|w(\alpha) - w_\rho\|^2 \, \rho_\mathcal{A}(\alpha) \, d\alpha. \tag{39}$$

We now observe that the second term above can be rewritten as follows

$$
\begin{aligned}
\int \|w(\alpha) - w_\rho\|^2 \, \rho_\mathcal{A}(\alpha) \, d\alpha \\
&= \int \|w(\alpha)\|^2 \, \rho_\mathcal{A}(\alpha) \, d\alpha - \|w_\rho\|^2 \\
&= \int \|w(\alpha)\|^2 \, \rho_\mathcal{A}(\alpha) \, d\alpha - \left\| \int w(\alpha') \rho_\mathcal{A}(\alpha') \, d\alpha' \right\|^2 \\
&= \int \|w(\alpha)\|^2 \, \rho_\mathcal{A}(\alpha) \, d\alpha - \int \langle w(\alpha), w(\alpha') \rangle \, \rho_\mathcal{A}(\alpha)\rho_\mathcal{A}(\alpha') \, d\alpha \, d\alpha' \\
&= \int \Big( \|w(\alpha)\|^2 - \langle w(\alpha), w(\alpha') \rangle \Big) \rho_\mathcal{A}(\alpha)\rho_\mathcal{A}(\alpha') \, d\alpha \, d\alpha'.
\end{aligned}
\tag{40}
$$

But, since

$$\int \|w(\alpha)\|^2 \, \rho_\mathcal{A}(\alpha)\rho_\mathcal{A}(\alpha') \, d\alpha = \frac{1}{2}\int \Big( \|w(\alpha)\|^2 + \|w(\alpha')\|^2 \Big) \rho_\mathcal{A}(\alpha')\rho_\mathcal{A}(\alpha') \, d\alpha \, d\alpha', \tag{41}$$

we conclude

$$\int \|w(\alpha) - w_\rho\|^2 \, \rho_\mathcal{A}(\alpha) \, d\alpha = \frac{1}{2}\int \|w(\alpha) - w(\alpha')\|^2 \, \rho_\mathcal{A}(\alpha)\rho_\mathcal{A}(\alpha') \, d\alpha \, d\alpha'. \tag{42}$$

Hence, substituting in (39), we get

$$\mathrm{Var}_\rho(w_\rho)^2 = 2\int \sigma(\alpha)^2 \, \rho_\mathcal{A}(\alpha) \, d\alpha + \frac{1}{2}\int \|w(\alpha) - w(\alpha')\|^2 \, \rho_\mathcal{A}(\alpha)\rho_\mathcal{A}(\alpha') \, d\alpha \, d\alpha'. \tag{43}$$

**Conditional variance.** We now focus on the conditional variance. As explained in Ex. 1, also in this case, we consider as side information a set of new features $X = (x_i)_{i=1}^n \in \cup_{n\in\mathbb{N}}\mathcal{X}^n$. As a consequence, we focus on conditioning functions of the form $\tau : \cup_{n\in\mathbb{N}}\mathcal{X}^n \to \mathbb{R}^d$. From Lemma 2, we know that the ideal function $\tau_\rho : \cup_{n\in\mathbb{N}}\mathcal{X}^n \to \mathbb{R}^d$ minimizing the conditional variance term over the space $\mathcal{T}$ of the measurable functions, is characterized, for almost every $X \in \cup_{n\in\mathbb{N}}\mathcal{X}^n$, by

$$\tau_\rho(X) = \mathbb{E}_{\mu\sim\rho(\cdot|X)} \, w_\mu = \int w_\mu \, \rho(\mu|X) \, dw_\mu. \tag{44}$$

We now observe that thanks to the parametrization of the environment $\rho$, for any target weight vector $w_\mu$ and features' set $X$, we can write

$$
\begin{aligned}
\rho(\mu|X) &= \frac{\rho(\mu,X)}{\rho_\mathcal{X}(X)} = \frac{\int \rho(\mu,X,\alpha) \, d\alpha}{\rho_\mathcal{X}(X)} = \int \rho(\mu|X,\alpha)\frac{\rho(X,\alpha)}{\rho_\mathcal{X}(X)} \, d\alpha \\
&= \int \rho(\mu|X,\alpha)\rho(\alpha|X) \, d\alpha = \int \rho(\mu|\alpha)\rho(\alpha|X) \, d\alpha,
\end{aligned}
\tag{45}
$$

where, in the last equality, we have exploited the fact that, by construction, $\mu$ is conditionally independent to $X$ w.r.t. $\alpha$, namely $\rho(\mu|X,\alpha) = \rho(\mu|\alpha)$. Then, substituting in (44), we get

$$
\begin{aligned}
\tau_\rho(X) &= \int w_\mu \, \rho(\mu|X) \, dw_\mu \\
&= \int w_\mu \left( \int \rho(\mu|\alpha)\rho(\alpha|X) \, d\alpha \right) dw_\mu \\
&= \int \left( \int w_\mu \, \rho(\mu|\alpha) \, dw_\mu \right) \rho(\alpha|X) \, d\alpha \\
&= \int w(\alpha) \, \rho(\alpha|X) \, d\alpha.
\end{aligned}
\tag{46}
$$

**Remark 5** (Asm. 2 in this example). *From the expression above, we can conclude that the function $\tau_\rho$ in (46) is a smooth function of $X$, if $\rho(X|\alpha)$ is a smooth function of $X$ for any $\alpha \in \mathcal{A}$. This means that, in such a case, there exist a Reproducing Kernel Hilbert Space (RKHS) $\mathcal{H}$ such that $\tau_\rho \in \mathcal{H}$ and, consequently, making Asm. 2 satisfied. For instance, we can take $\mathcal{H}$ to be the space induced by the Abel kernel*

$$k(X, X') = e^{-\sum_{j=1}^{n} \frac{\|x_j - x_j'\|}{\sigma}}, \qquad \sigma > 0 \qquad X, X' \in \cup_{n \in \mathbb{N}} \mathcal{X}^n. \tag{47}$$

*In this case, $\mathcal{H} = W^{d/2+1,2}$ corresponds the Sobolev's space of functions with square integrable $d/2 + 1$ derivatives.*

We now proceed by computing the conditional variance. In order to do this, we observe that

$$\mathrm{Var}_\rho(\tau_\rho)^2 = \mathbb{E}_{(\mu,X)\sim\rho} \left\| w_\mu - \tau_\rho(X) \right\|^2 = \mathbb{E}_{X\sim\rho_\mathcal{X}} \, \mathbb{E}_{\mu\sim\rho(\cdot|X)} \left\| w_\mu - \tau_\rho(X) \right\|^2. \tag{48}$$

We now observe that, for any set of features $X$, exploiting (45), we can rewrite the inner expectation above as follows

$$
\begin{aligned}
\mathbb{E}_{\mu\sim\rho(\cdot|X)} \left\| w_\mu - \tau_\rho(X) \right\|^2 &= \int \left\| w_\mu - \tau_\rho(X) \right\|^2 \rho(\mu|X) \, dw_\mu \\
&= \int \left\| w_\mu - \tau_\rho(X) \right\|^2 \left( \int \rho(\mu|\alpha)\rho(\alpha|X) \, d\alpha \right) dw_\mu \\
&= \int \left( \int \left\| w_\mu - \tau_\rho(X) \right\|^2 \rho(\mu|\alpha) \, dw_\mu \right) \rho(\alpha|X) \, d\alpha.
\end{aligned}
\tag{49}
$$

But, for each $\alpha \in \mathcal{A}$, we can write

$$
\begin{aligned}
&\int \left\| w_\mu - \tau_\rho(X) \right\|^2 \rho(\mu|\alpha) \, dw_\mu \\
&= \int \left\| w_\mu \right\|^2 \rho(\mu|\alpha) \, dw_\mu - 2 \left\langle w(\alpha), \tau_\rho(X) \right\rangle + \left\| \tau_\rho(X) \right\|^2 \\
&= \int \left\| w_\mu \right\|^2 \rho(\mu|\alpha) \, dw_\mu \pm \left\| w(\alpha) \right\|^2 - 2 \left\langle w(\alpha), \tau_\rho(X) \right\rangle + \left\| \tau_\rho(X) \right\|^2 \\
&= 2\sigma(\alpha)^2 + \left\| w(\alpha) - \tau_\rho(X) \right\|^2.
\end{aligned}
\tag{50}
$$

Hence, substituting into (49), we get

$$\mathbb{E}_{\mu\sim\rho(\cdot|X)} \left\| w_\mu - \tau_\rho(X) \right\|^2 = 2 \int \sigma(\alpha)^2 \rho(\alpha|X) \, d\alpha + \int \left\| w(\alpha) - \tau_\rho(X) \right\|^2 \rho(\alpha|X) \, d\alpha. \tag{51}$$

Hence, integrating w.r.t. $X$, we get

$$
\begin{aligned}
\mathrm{Var}_\rho(\tau_\rho)^2 &= \mathbb{E}_{X\sim\rho_\mathcal{X}} \, \mathbb{E}_{\mu\sim\rho(\cdot|X)} \left\| w_\mu - \tau_\rho(X) \right\|^2 \\
&= 2 \int \sigma(\alpha)^2 \rho(\alpha|X)\rho_\mathcal{X}(X) \, d\alpha \, dX + \int \left\| w(\alpha) - \tau_\rho(X) \right\|^2 \rho(\alpha|X)\rho_\mathcal{X}(X) \, d\alpha \, dX \\
&= 2 \int \sigma(\alpha)^2 \rho_\mathcal{A}(\alpha) \, d\alpha + \int \left\| w(\alpha) - \tau_\rho(X) \right\|^2 \rho(\alpha|X)\rho_\mathcal{X}(X) \, d\alpha \, dX.
\end{aligned}
\tag{52}
$$

We now observe that, exploiting the closed form of $\tau_\rho$ in (46), the second term above can be rewritten as follows

$$
\begin{aligned}
&\int \left\| w(\alpha) - \tau_\rho(X) \right\|^2 \rho(\alpha|X)\rho_\mathcal{X}(X) \, d\alpha \, dX \\
&= \int \left\| w(\alpha) - \int w(\alpha') \, \rho(\alpha'|X) \, d\alpha' \right\|^2 \rho(\alpha|X)\rho_\mathcal{X}(X) \, d\alpha \, dX \\
&= \int \left\| w(\alpha) \right\|^2 \rho(\alpha|X)\rho_\mathcal{X}(X) \, d\alpha \, dX - 2 \int \left\langle w(\alpha), w(\alpha') \right\rangle \rho(\alpha|X)\rho(\alpha'|X)\rho_\mathcal{X}(X) \, d\alpha \, d\alpha' \, dX \\
&\quad + \int \left\| \int w(\alpha') \, \rho(\alpha'|X) \, d\alpha' \right\|^2 \rho_\mathcal{X}(X) \, dX.
\end{aligned}
\tag{53}
$$

Note now that

$$\int \|w(\alpha)\|^2 \; \rho(\alpha|X)\rho_{\mathcal{X}}(X) \, d\alpha \, dX$$

$$= \frac{1}{2}\left( \int \|w(\alpha)\|^2 \; \rho(\alpha|X)\rho_{\mathcal{X}}(X) \, d\alpha \, dX + \int \|w(\alpha')\|^2 \; \rho(\alpha'|X)\rho_{\mathcal{X}}(X) \, d\alpha' \, dX \right) \quad (54)$$

$$= \frac{1}{2}\left( \int \Big( \|w(\alpha)\|^2 + \|w(\alpha')\|^2 \Big) \rho(\alpha|X)\rho(\alpha'|X)\rho_{\mathcal{X}}(X) \, d\alpha \, d\alpha' \, dX \right)$$

and

$$\int \left\|\int w(\alpha') \, \rho(\alpha'|X) \, d\alpha' \right\|^2 \rho_{\mathcal{X}}(X) \, dX = \int \langle w(\alpha), w(\alpha') \rangle \; \rho(\alpha|X)\rho(\alpha'|X)\rho_{\mathcal{X}}(X) \, d\alpha \, d\alpha' \, dX. \quad (55)$$

Substituting (54) and (55) in (53), we get

$$\int \|w(\alpha) - \tau_\rho(X)\|^2 \; \rho(\alpha|X)\rho_{\mathcal{X}}(X) \, d\alpha \, dX$$

$$= \int \frac{1}{2}\Big( \|w(\alpha)\|^2 - 2 \langle w(\alpha), w(\alpha') \rangle + \|w(\alpha')\|^2 \Big) \rho(\alpha|X)\rho(\alpha'|X)\rho_{\mathcal{X}}(X) \, d\alpha \, d\alpha' \, dX$$

$$= \frac{1}{2} \int \|w(\alpha) - w(\alpha')\|^2 \rho(\alpha|X)\rho(\alpha'|X)\rho_{\mathcal{X}}(X) \, d\alpha \, d\alpha' \, dX \quad (56)$$

$$= \frac{1}{2} \int \|w(\alpha) - w(\alpha')\|^2 \frac{\rho(X|\alpha)\rho(X|\alpha')}{\rho_{\mathcal{X}}(X)} \, \rho_{\mathcal{A}}(\alpha)\rho_{\mathcal{A}}(\alpha') \, d\alpha \, d\alpha' \, dX$$

$$= \frac{1}{2} \int \|w(\alpha) - w(\alpha')\|^2 \left( \int \frac{\rho(X|\alpha)\rho(X|\alpha')}{\rho_{\mathcal{X}}(X)} \, dX \right)\rho_{\mathcal{A}}(\alpha)\rho_{\mathcal{A}}(\alpha') \, d\alpha \, d\alpha'.$$

Hence, the conditional variance is given by

$$\mathrm{Var}_\rho(\tau_\rho)^2 = 2 \int \sigma(\alpha)^2 \, \rho(\alpha) \, d\alpha$$

$$+ \frac{1}{2} \int \|w(\alpha) - w(\alpha')\|^2 \left( \int \frac{\rho(X|\alpha)\rho(X|\alpha')}{\rho_{\mathcal{X}}(X)} \, dX \right)\rho_{\mathcal{A}}(\alpha)\rho_{\mathcal{A}}(\alpha') \, d\alpha \, d\alpha'. \quad (57)$$

**Conditional vs unconditional variance.** Subtracting (57) to (43), we get that the difference between the unconditional and conditional variance is given by the following closed form

$$\mathrm{Var}_\rho(w_\rho)^2 - \mathrm{Var}_\rho(\tau_\rho)^2$$

$$= \frac{1}{2} \int \left( 1 - \int \frac{\rho(X|\alpha)\rho(X|\alpha')}{\rho_{\mathcal{X}}(X)} \, dX \right) \|w(\alpha) - w(\alpha')\|^2 \, \rho_{\mathcal{A}}(\alpha)\rho_{\mathcal{A}}(\alpha') \, d\alpha \, d\alpha'. \quad (58)$$

Hence, if

$$\int \frac{\rho(X|\alpha)\rho(X|\alpha')}{\rho_{\mathcal{X}}(X)} \, dX \leq \epsilon(\alpha, \alpha') \quad (59)$$

for some $\epsilon : \mathcal{A} \times \mathcal{A} \to \mathbb{R}_+$, we can write

$$\mathrm{Var}_\rho(w_\rho)^2 - \mathrm{Var}_\rho(\tau_\rho)^2 \geq \frac{1}{2} \int \Big( 1 - \epsilon(\alpha, \alpha') \Big) \|w(\alpha) - w(\alpha')\|^2 \, \rho_{\mathcal{A}}(\alpha)\rho_{\mathcal{A}}(\alpha') \, d\alpha \, d\alpha'. \quad (60)$$

### B.2 Clusters (Ex. 1)

The example in the section above encompasses the setting outlined in Ex. 1, by identifying the latent variable $\alpha$ with the clusters' indexes, namely, $\mathcal{A} = \{1, \ldots, m\}$ and, for any $\alpha \in \mathcal{A}$, $\rho_{\mathcal{A}}(\alpha) = 1/m$. We now show that adapting the results above to this specific setting, we manage to show the statement in Ex. 1 in the main body.

**Unconditional variance.** Specifying (43) to the setting outlined in Ex. 1, we get the following closed form for the unconditional variance:

$$\text{Var}_\rho(w_\rho)^2 = \frac{2}{m} \sum_{\alpha=1}^m \sigma(\alpha)^2 + \frac{1}{2m^2} \sum_{\alpha,\alpha'=1}^m \|w(\alpha) - w(\alpha')\|^2. \tag{61}$$

**Conditional variance.** Specifying (57) to the setting outlined in Ex. 1, we get the following closed form for the conditional variance:

$$\text{Var}_\rho(\tau_\rho)^2 = \frac{2}{m} \sum_{\alpha=1}^m \sigma(\alpha)^2 + \frac{1}{2m^2} \sum_{\alpha,\alpha'=1}^m \left( \int \frac{\rho(X|\alpha)\rho(X|\alpha')}{\rho_{\mathcal{X}}(X)} \, dX \right) \|w(\alpha) - w(\alpha')\|^2. \tag{62}$$

**Conditional vs unconditional variance.** Finally, specifying (58) to the setting outlined in Ex. 1, we get the following closed form for the gap between the unconditional and the conditional variance:

$$\text{Var}_\rho(w_\rho)^2 - \text{Var}_\rho(\tau_\rho)^2 = \frac{1}{2m^2} \sum_{\alpha,\alpha'=1}^m \left( 1 - \int \frac{\rho(X|\alpha)\rho(X|\alpha')}{\rho_{\mathcal{X}}(X)} \, dX \right) \|w(\alpha) - w(\alpha')\|^2. \tag{63}$$

The last ingredient we need to prove the upper bound in Ex. 1 is the following.

**Proposition 6.** *Assume now that for any $\alpha \in \mathcal{A} = \{1, \ldots, m\}$, $\rho(X|\alpha)$ is a Gaussian distribution with mean $x(\alpha) \in \mathbb{R}^d$ and variance $\sigma_{\mathcal{X}}^2$. Then, for any $\alpha, \alpha' \in \mathcal{A}$,*

$$\int \frac{\rho(X|\alpha)\rho(X|\alpha')}{\rho_{\mathcal{X}}(X)} \, dX \leq \frac{m}{2} \, e^{-\frac{n}{\sigma_{\mathcal{X}}^2} \|x(\alpha)-x(\alpha')\|^2}. \tag{64}$$

**Proof.** Thanks to the composition of the environment in clusters, we can write

$$\rho(X) = \sum_{\epsilon=1}^m \rho(X|\epsilon)\rho_{\mathcal{A}}(\epsilon) = \frac{1}{m} \sum_{\epsilon=1}^m \rho(X|\epsilon). \tag{65}$$

As a consequence, for any $\alpha, \alpha' \in \mathcal{A}$, we can write

$$\begin{aligned}
\int \frac{\rho(X|\alpha)\rho(X|\alpha')}{\rho_{\mathcal{X}}(X)} \, dX &= m \int \frac{\rho(X|\alpha)\rho(X|\alpha')}{\sum_{\epsilon=1}^m \rho(X|\epsilon)} \, dX \\
&\leq m \int \frac{\rho(X|\alpha)\rho(X|\alpha')}{\rho(X|\alpha) + \rho(X|\alpha')} \, dX \\
&\leq \frac{m}{2} \int \sqrt{\rho(X|\alpha)\rho(X|\alpha')} \, dX,
\end{aligned} \tag{66}$$

where in the last inequality we have used the inequality

$$\frac{ab}{a+b} \leq \frac{\sqrt{ab}}{2}, \tag{67}$$

holding for any $a, b > 0$. We now observe that, by assumption, we are considering Gaussian distributions for the inputs' probability, i.e., for any $\alpha \in \{1, \ldots, k\}$, we have

$$\rho(X|\alpha) = \prod_{j=1}^n \frac{1}{\sqrt{2\pi\sigma_{\mathcal{X}}^2}} \, e^{-\frac{\|x_j - x(\alpha)\|^2}{\sigma_{\mathcal{X}}^2}}. \tag{68}$$

Hence, we have

$$\int \sqrt{\rho(X|\alpha)\rho(X|\alpha')} \, dX = \frac{1}{\Pi_{j=1}^n \sqrt{2\pi\sigma_{\mathcal{X}}^2}} \int e^{-\frac{1}{\sigma_{\mathcal{X}}^2} \sum_{j=1}^n \|x_j - x(\alpha)\|^2 + \|x_j - x(\alpha')\|^2} \Pi_j dx_j. \tag{69}$$

We now observe that

$$\|x_j - x(\alpha)\|^2 + \|x_j - x(\alpha')\|^2$$

$$= 2\|x_j\|^2 - 2\langle x_j, x(\alpha) + x(\alpha')\rangle + \|x(\alpha)\|^2 + \|x(\alpha')\|^2$$

$$= 2\|x_j\|^2 - 2\langle x_j, x(\alpha) + x(\alpha')\rangle + \|x(\alpha)\|^2 + \|x(\alpha')\|^2 \pm \frac{1}{2}\|x(\alpha) + x(\alpha')\|^2$$

$$= \left\|\sqrt{2}x_j - \frac{1}{\sqrt{2}}(x(\alpha) + x(\alpha'))\right\|^2 - \frac{1}{2}\|x(\alpha) + x(\alpha')\|^2 + \|x(\alpha)\|^2 + \|x(\alpha')\|^2 \qquad (70)$$

$$= \left\|\sqrt{2}x_j - \frac{1}{\sqrt{2}}(x(\alpha) + x(\alpha'))\right\|^2 + \frac{1}{2}\|x(\alpha)\|^2 + \frac{1}{2}\|x(\alpha')\|^2 - \langle x(\alpha), x(\alpha')\rangle$$

$$= \left\|\sqrt{2}x_j - \frac{1}{\sqrt{2}}(x(\alpha) + x(\alpha'))\right\|^2 + \frac{1}{2}\|x(\alpha) - x(\alpha')\|^2.$$

Substituting (70) into (69), we conclude

$$\int \sqrt{\rho(X|\alpha)\rho(X|\alpha')}\, dX \leq$$

$$\leq e^{-\frac{n}{\sigma_{\mathcal{X}}^2}\|x(\alpha) - x(\alpha')\|^2} \frac{1}{\Pi_{j=1}^n \sqrt{2\pi\sigma_{\mathcal{X}}^2}} \int e^{-\frac{1}{\sigma_{\mathcal{X}}^2}\sum_{j=1}^n \left\|\sqrt{2}x_j - \frac{1}{\sqrt{2}}(x(\alpha) + x(\alpha'))\right\|^2} \Pi_i dx_i$$

$$= e^{-\frac{n}{\sigma_{\mathcal{X}}^2}\|x(\alpha) - x(\alpha')\|^2} \frac{1}{\Pi_{j=1}^n \sqrt{2\pi\sigma_{\mathcal{X}}^2}} \int e^{-\frac{1}{\sigma_{\mathcal{X}}^2}\sum_{j=1}^n \left\|x_j - \frac{x(\alpha)+x(\alpha')}{2}\right\|^2} \Pi_i dx_i \qquad (71)$$

$$= e^{-\frac{n}{\sigma_{\mathcal{X}}^2}\|x(\alpha) - x(\alpha')\|^2},$$

where in the last equality we have exploited the integral of the Gaussian distribution $\mathcal{N}\left(\frac{x(\alpha)+x(\alpha')}{2}, \sigma_{\mathcal{X}}\right)$:

$$\frac{1}{\Pi_{j=1}^n \sqrt{\pi\sigma_{\mathcal{X}}^2}} \int e^{-\frac{1}{2\sigma_{\mathcal{X}}^2}\sum_{j=1}^n \left\|x_j - \frac{x(\alpha)+x(\alpha')}{2}\right\|^2} \Pi_i dx_i = 1. \qquad (72)$$

Using the last inequality above in (66), we get the desired statement. ∎

The desired statement in Ex. 1 derives from combining (63) with (64).

## B.3 Circle (Ex. 2)

Consider now the setting outlined in Ex. 2. We proceed as before: we first compute the unconditional variance, then, the conditional variance and, finally, the gap between them.

**Unconditional variance.** We start from observing that, since by construction, for any $s \in [0,1]$, $\rho(\mu|s)$ is the Gaussian distribution with mean $h(s)$, $\rho_{\mathcal{S}}$ is the uniform distribution on $[0,1]$ and $h$ is centered in $c$, then, we have

$$w_\rho = \mathbb{E}_{\mu\sim\rho}\, w_\mu = \mathbb{E}_{s\sim\rho_{\mathcal{S}}}\, \mathbb{E}_{\mu\sim\rho(\cdot|s)}\, w_\mu = \mathbb{E}_{s\sim\rho_{\mathcal{S}}}\, h(s) = c. \qquad (73)$$

Hence, we can rewrite the unconditional variance as follows

$$\mathrm{Var}_\rho(w_\rho)^2 = \mathbb{E}_{(\mu,s)\sim\rho}\|w_\mu - c\|^2$$

$$= \int \|w_\mu - c \pm h(s)\|^2\, \rho(\mu,s)\, dw_\mu ds$$

$$= \int \left(\int \|w_\mu - h(s)\|^2\, \rho(\mu|s)\, dw_\mu\right) \rho_{\mathcal{S}}(s)\, ds + \int \|h(s) - c\|^2\, \rho_{\mathcal{S}}(s)\, ds \qquad (74)$$

$$+ \int \left\langle c - h(s), \int (w_\mu - h(s))\, \rho(\mu|s)\, dw_\mu\right\rangle \rho_{\mathcal{S}}(s)\, ds$$

$$= \sigma^2 + r^2,$$

where, in the last equality, we have exploited the fact $\|h(s) - c\| = r$ for any $s \in \mathcal{S}$ and the fact that, thanks to the assumption $\rho(\cdot|s) = \mathcal{N}(h(s), \sigma^2 I)$,

$$\int (w_\mu - h(s))\, \rho(\mu|s)\, dw_\mu = 0 \qquad \int \|w_\mu - h(s)\|^2\, \rho(\mu|s)\, dw_\mu = \sigma^2. \tag{75}$$

**Conditional variance.** Since by construction $\rho(\cdot|s) = \mathcal{N}(h(s), \sigma^2 I)$, we immediately see that the ideal function $\tau_\rho : [0,1] \to \mathbb{R}^d$ in Lemma 2 and the corresponding conditional variance can be, respectively, rewritten as follows

$$\tau_\rho(s) = \mathbb{E}_{\mu\sim\rho(\cdot|s)}\, w_\mu = h(s) \tag{76}$$

$$\mathrm{Var}_\rho(\tau_\rho)^2 = \mathbb{E}_{(w_\mu,s)\sim\rho}\, \|w_\mu - h(s)\|^2 = \int \left( \int \|w_\mu - h(s)\|^2\, \rho(\mu|s)\, dw_\mu \right) \rho_\mathcal{S}\, ds = \sigma^2. \tag{77}$$

**Conditional vs unconditional variance.** Subtracting (77) to (74), we get that the difference between the unconditional and conditional variance is given by

$$\mathrm{Var}_\rho(w_\rho)^2 - \mathrm{Var}_\rho(\tau_\rho)^2 = r^2. \tag{78}$$

All the statements given in Ex. 2 have hence been proven.

## C   Closed forms for Asm. 2

Thanks to Asm. 2, we know that there exist $M_\rho \in \mathbb{R}^{d\times k}$ and $b_\rho \in \mathbb{R}^d$ such that $\tau_\rho(\cdot) = M_\rho \Phi(\cdot) + b_\rho$. In the following lemma, we give the closed form of these quantities and the corresponding variance. We let $\mathrm{Tr}(\cdot)$ and $\cdot^*$ be the trace and the conjugate operators respectively.

**Lemma 7** (Best linear conditioning function in hindsight). *Recall the vector $w_\rho = \mathbb{E}_{\mu\sim\rho_\mathcal{M}}\, w_\mu$ and introduce the vector $\nu_\rho = \mathbb{E}_{s\sim\rho_\mathcal{S}}\, \Phi(s)$. Introduce also the following covariance matrices*

$$\mathrm{Cov}_\rho(s,s) = \mathbb{E}_{s\sim\rho_\mathcal{S}} \left[ (\Phi(s) - \nu_\rho)(\Phi(s) - \nu_\rho)^\top \right] \in \mathbb{R}^{k\times k} \tag{79}$$

$$\mathrm{Cov}_\rho(w,w) = \mathbb{E}_{\mu\sim\rho_\mathcal{M}} \left[ (w_\mu - w_\rho)(w_\mu - w_\rho)^\top \right] \in \mathbb{R}^{d\times d} \tag{80}$$

$$\mathrm{Cov}_\rho(w,s) = \mathbb{E}_{(\mu,s)\sim\rho} \left[ (w_\mu - w_\rho)(\Phi(s) - \nu_\rho)^\top \right] \in \mathbb{R}^{d\times k}. \tag{81}$$

*Then,*

$$\min_{M\in\mathbb{R}^{d\times k}, b\in\mathbb{R}^d} \mathrm{Var}_\rho(\tau_{M,b})^2 = \mathrm{Var}_\rho(w_\rho)^2 - \mathrm{Tr}\Big( \mathrm{Cov}_\rho(w,w)\mathrm{Corr}_\rho(w,s)^\top \mathrm{Corr}_\rho(w,s) \Big)$$
$$= \mathrm{Var}_\rho(w_\rho)^2 - \big\| \mathrm{Cov}_\rho(s,s)^{1/2} M_\rho \big\|_F^2, \tag{82}$$

*where we have introduced the correlation matrix*

$$\mathrm{Corr}_\rho(w,s) = \mathrm{Cov}_\rho(w,w)^{\dagger/2}\, \mathrm{Cov}_\rho(w,s)\mathrm{Cov}_\rho(s,s)^{\dagger/2} \in \mathbb{R}^{d\times k}. \tag{83}$$

*Moreover, the (minimum norm) values at which the minimum above is attained are given by*

$$M_\rho = \mathrm{Cov}_\rho(w,s)\mathrm{Cov}_\rho(s,s)^\dagger \tag{84}$$

$$b_\rho = w_\rho - \mathrm{Cov}_\rho(w,s)\mathrm{Cov}_\rho(s,s)^\dagger\, \nu_\rho. \tag{85}$$

When Asm. 2 holds, the minimum conditional variance in Lemma 2 can be rewritten as $\min_{\tau\in\mathcal{T}} \mathrm{Var}_\rho(\tau)^2 = \min_{\tau\in\mathcal{T}_\Phi} \mathrm{Var}_\rho(\tau)^2$. As a consequence, in this case, the statement above in (82) allows us to express the gap between the conditional and the unconditional variance in (12) as a function of the correlation between the target tasks' weight vectors and the side information. In addition, we can also deduce that such a gap is significant when the 'inclination' of the

linear relation linking the target tasks' weight vectors and the side information (more formally, $\|\mathrm{Cov}_\rho(s,s)^{1/2}M_\rho\|_F^2$) is large. This is not surprising, since, in this case, the gap between conditional and unconditional meta-learning can be interpreted as the gap in using the best linear function w.r.t. the constant one $\tau \equiv w_\rho$.

As we will see in the following, the proof of Lemma 7, directly derives from the following facts regarding linear Least Squares.

**Lemma 8.** *Let $\mathcal{X}$ be an Hilbert space, $\mathcal{Y} = \mathbb{R}^d$ and $\mathcal{H} = \mathbb{R}^k$. Consider a map $\Psi : \mathcal{X} \to \mathcal{H}$ and a joint probability distribution $\rho$ on $\mathcal{X} \times \mathcal{Y}$ with conditional distribution $\rho(x|y)$ and marginal $\rho_{\mathcal{Y}}(y)$. Denote by $\otimes$ the standard outer product, introduce the covariance operators:*

$$C_{yy} = \mathbb{E}\left[y \otimes y\right] \qquad C_{xx} = \mathbb{E}\left[\Psi(x) \otimes \Psi(x)\right] \qquad C_{xy} = \mathbb{E}\left[y \otimes \Psi(x)\right] \tag{86}$$

*and the correlation operator*

$$\mathrm{Corr}_{xy} = C_{xx}^{\dagger/2} C_{xy} C_{yy}^{\dagger/2}. \tag{87}$$

*Then,*

$$\min_{M \in \mathbb{R}^{d \times k}} \mathbb{E}_{(x,y) \sim \rho} \left\|y - M\Psi(x)\right\|^2 = \mathrm{Tr}(C_{yy}) - \mathrm{Tr}\left(C_{yy}\mathrm{Corr}_{xy}^* \mathrm{Corr}_{xy}\right)$$

$$= \mathrm{Tr}(C_{yy}) - \left\|C_{xx}^{1/2}M_\rho\right\|_F^2. \tag{88}$$

*The optimal (minimum norm) matrix $M_\rho$ is given by*

$$M_\rho = C_{xy}C_{xx}^{\dagger}. \tag{89}$$

**Proof.** For any $M \in \mathbb{R}^{d \times k}$, we can rewrite

$$\mathbb{E}_{(x,y) \sim \rho} \left\|y - M\Psi(x)\right\|^2 = \mathbb{E}_{y \sim \rho_{\mathcal{Y}}} \|y\|^2 + \mathbb{E}_{x \sim \rho_{\mathcal{X}}} \left\|M\Psi(x)\right\|^2 - 2\,\mathbb{E}_{(x,y) \sim \rho} \left\langle y, M\Psi(x)\right\rangle$$

$$= \mathrm{Tr}(C_{yy}) + \mathrm{Tr}(C_{xx}M^*M) - 2\mathrm{Tr}(C_{xy}^*M). \tag{90}$$

By setting the derivatives w.r.t. $M$ equal to zero, we know that the optimal matrix $M_\rho$ satisfies

$$M_\rho C_{xx} = C_{xy}. \tag{91}$$

Hence, the optimal (minimum norm) matrix $M_\rho$ is given by

$$M_\rho = C_{xy}C_{xx}^{\dagger}. \tag{92}$$

We now compute the corresponding minimum value. We first observe that, by the closed form of the optimal matrix $M_\rho$, we can rewrite

$$\mathrm{Tr}(C_{xx}M_\rho^*M_\rho) = \mathrm{Tr}(C_{xx}C_{xx}^{\dagger}C_{xy}^*C_{xy}C_{xx}^{\dagger}) = \mathrm{Tr}(C_{xx}^{\dagger}C_{xx}C_{xx}^{\dagger}C_{xy}^*C_{xy})$$

$$= \mathrm{Tr}(C_{xx}^{\dagger}C_{xy}^*C_{xy}), \tag{93}$$

where in the last equality we have applied the identity $C_{xx}^{\dagger}C_{xx}C_{xx}^{\dagger} = C_{xx}^{\dagger}$. We then observe that, again, by the closed form of the optimal matrix $M_\rho$, we can rewrite

$$\mathrm{Tr}(C_{xy}^*M_\rho) = \mathrm{Tr}(C_{xy}^*C_{xy}C_{xx}^{\dagger}) = \mathrm{Tr}(C_{xx}^{\dagger}C_{xy}^*C_{xy}). \tag{94}$$

Substituting (93) and (94) in (90), we get the following:

$$\min_{M \in \mathbb{R}^{d \times k}} \mathbb{E}_{(x,y) \sim \rho} \left\|y - M\Psi(x)\right\|^2 = \mathrm{Tr}(C_{yy}) - \mathrm{Tr}(C_{xx}^{\dagger}C_{xy}^*C_{xy}) = \mathrm{Tr}(C_{yy}) - \left\|C_{xx}^{1/2}M_\rho\right\|_F^2, \tag{95}$$

where in the last equality we have applied the optimality condition (91). In order to terminate the proof, we need to prove the following equality

$$\mathrm{Tr}(C_{xx}^{\dagger}C_{xy}^*C_{xy}) = \mathrm{Tr}(C_{yy}\mathrm{Corr}_{xy}^*\mathrm{Corr}_{xy}). \tag{96}$$

In order to do this, we proceed as follows. Let $L^2(\mathcal{Y}, \mathbb{R}, \rho_{\mathcal{Y}})$ the space of functions from $\mathcal{Y}$ to $\mathbb{R}$ that are square integrable w.r.t. $\rho_{\mathcal{Y}}$ and recall that, for any $f, g \in L^2(\mathcal{Y}, \mathbb{R}, \rho_{\mathcal{Y}})$, such a space is endowed with the scalar product

$$\langle f, g \rangle_{L^2} = \int f(y)g(y)\, d\rho_{\mathcal{Y}}(y). \tag{97}$$

Throughout the rest of the proof we will use the following operator

$$S : \mathcal{Y} \to L^2(\mathcal{Y}, \mathbb{R}, \rho_{\mathcal{Y}}) \qquad h \mapsto (y \mapsto \langle h, \cdot \rangle_{\mathcal{Y}}), \tag{98}$$

where $\langle \cdot, \cdot \rangle_{\mathcal{Y}}$ is the scalar product in $\mathcal{Y}$. Its adjoint operator $S^* : L^2(\mathcal{Y}, \mathbb{R}, \rho_{\mathcal{Y}}) \to \mathcal{Y}$ is such that, for any $h \in \mathcal{Y}$ and function $f \in L^2(\mathcal{Y}, \mathbb{R}, \rho_{\mathcal{Y}})$,

$$\langle h, S^* f \rangle_{\mathcal{Y}} = \langle Sh, f \rangle_{L^2} = \int f(y) \langle h, y \rangle_{\mathcal{Y}} \, d\rho_{\mathcal{Y}}(y) = \left\langle h, \int y f(y) \, d\rho_{\mathcal{Y}}(y) \right\rangle_{\mathcal{Y}}. \tag{99}$$

This implies that, for any $f \in L^2(\mathcal{Y}, \mathbb{R}, \rho_{\mathcal{Y}})$,

$$S^* f = \int y f(y) \, d\rho_{\mathcal{Y}}(y). \tag{100}$$

In order to prove the desired statement in (96), we will use the two facts below.

**First fact.** The first fact we need is to show that the operator $S^* S$ coincides with the covariance operator $C_{yy}$, i.e.

$$S^* S = C_{yy} \qquad C_{yy} = \mathbb{E}[y \otimes y]. \tag{101}$$

This fact holds, as a matter of fact, we immediately see that, for any $h_1, h_2 \in \mathcal{Y}$, we can write

$$\langle h_1, S^* S h_2 \rangle_{\mathcal{Y}} = \langle Sh_1, Sh_2 \rangle_{L^2} = \int \langle h_1, y \rangle_{\mathcal{Y}} \langle h_2, y \rangle_{\mathcal{Y}} \, d\rho_{\mathcal{Y}}(y)$$

$$= \left\langle h_1, \left( \int y \otimes y \, d\rho_{\mathcal{Y}}(y) \right) h_2 \right\rangle_{\mathcal{Y}} = \langle h_1, C_{yy} h_2 \rangle_{\mathcal{Y}}. \tag{102}$$

**Second fact.** Now, recall the map $\Psi : \mathcal{X} \to \mathcal{H}$ in the statement and define $G : \mathcal{Y} \to \mathcal{H}$ the function

$$G(y) = \int \Psi(x) \, d\rho(x|y) \tag{103}$$

mapping $y$ into the conditional expectation of $\rho(x|y)$. Assume that $G \in L^2(\mathcal{Y}, \mathcal{H}, \rho_{\mathcal{Y}})$, the space of functions from $\mathcal{Y}$ to $\mathcal{H}$ that are square integrable w.r.t. $\rho_{\mathcal{Y}}$. Note that $L^2(\mathcal{Y}, \mathcal{H}, \rho_{\mathcal{Y}})$ is isometric to $\mathcal{H} \otimes L^2(\mathcal{Y}, \mathbb{R}, \rho_{\mathcal{Y}})$. Denote $J : L^2(\mathcal{Y}, \mathcal{H}, \rho_{\mathcal{Y}}) \to \mathcal{H} \otimes L^2(\mathcal{Y}, \mathbb{R}, \rho_{\mathcal{Y}})$ such an isometry and let $J_G = J(G)$ the Hilbert-Schmidt operator from $L^2(\mathcal{Y}, \mathbb{R}, \rho_{\mathcal{Y}})$ to $\mathcal{H}$ associated to $G$. Recall that the isometry follows from the observation that, given a basis $\{h_i\}_{i \in \mathbb{N}}$ of $\mathcal{H}$ and $\{f_j\}_{j \in \mathbb{N}}$ of $L^2(\mathcal{Y}, \mathbb{R}, \rho_{\mathcal{Y}})$, then the sequence $\{J(g_{ij})\}_{i,j \in \mathbb{N}}$, with $g_{ij}$ the vector-valued functions $g_{ij}(y) = h_i f_j(y)$ and such that $J(g_{ij}) = h_i \otimes f_j$, forms a basis for $\mathcal{H} \otimes L^2(\mathcal{Y}, \mathbb{R}, \rho_{\mathcal{Y}})$.

By construction, denoting by $\langle \cdot, \cdot \rangle_{\mathcal{H}}$ and $\langle \cdot, \cdot \rangle_{\mathcal{H} \otimes L^2}$ the scalar product in $\mathcal{H}$ and $\mathcal{H} \otimes L^2(\mathcal{Y}, \mathbb{R}, \rho_{\mathcal{Y}})$ respectively, for any $f \in L^2(\mathcal{Y}, \mathbb{R}, \rho_{\mathcal{Y}})$ and $h \in \mathcal{H}$, we have

$$\langle J_G, h \otimes f \rangle_{\mathcal{H} \otimes L^2} = \langle G(\cdot), h f(\cdot) \rangle_{L^2(\mathcal{Y}, \mathcal{H}, \rho_{\mathcal{Y}})} = \int \langle G(y), h \rangle_{\mathcal{H}} f(y) \, d\rho_{\mathcal{Y}}(y). \tag{104}$$

The second fact we need is to show that the operator $J_G S$ coincides with the covariance operator $C_{xy}$, i.e.

$$J_G S = C_{xy} \qquad C_{xy} = \mathbb{E}[y \otimes \Psi(x)]. \tag{105}$$

Also this fact holds, as a matter of fact, for any $h_1 \in \mathcal{H}$ and $h_2 \in \mathcal{Y}$, we can write the following

$$\langle h_1, J_G S h_2 \rangle_{\mathcal{H}} = \langle J_G, h_1 \otimes (Sh_2) \rangle_{\mathcal{H} \otimes L^2}$$

$$= \int \langle G(y), h_1 \rangle_{\mathcal{H}} (Sh_2)(y) \, d\rho_{\mathcal{Y}}(y)$$

$$= \int \langle G(y), h_1 \rangle_{\mathcal{H}} \langle h_2, y \rangle_{\mathcal{Y}} \, d\rho_{\mathcal{Y}}(y)$$

$$= \int \langle h_1, (y \otimes G(y)) h_2 \rangle_{\mathcal{H}} \, d\rho_{\mathcal{Y}}(y) \tag{106}$$

$$= \left\langle h_1, \left( \int y \otimes G(y) \, d\rho_{\mathcal{Y}}(y) \right) h_2 \right\rangle_{\mathcal{H}}$$

$$= \langle h_1, C_{xy} h_2 \rangle_{\mathcal{H}},$$

where in the last inequality, we have exploited the definition of $G$ according to which

$$\int y \otimes G(y) \, d\rho_{\mathcal{Y}}(y) = \int y \otimes \left( \int \Psi(x) \, d\rho(x|y) \right) d\rho_{\mathcal{Y}}(y) = \int y \otimes \Psi(x) \, d\rho(x, y) = C_{xy}. \quad (107)$$

As a consequence, recalling the covariance operator $C_{xx} = \mathbb{E}\left[\Psi(x) \otimes \Psi(x)\right]$ and combining the two facts above, we can write the following steps:

$$\begin{aligned}
\mathrm{Tr}\big(C_{xx}^{\dagger} C_{xy}^{*} C_{xy}\big) &= \mathrm{Tr}\big(C_{xx}^{\dagger} J_G S S^* J_G^*\big) \\
&= \mathrm{Tr}\big(C_{xx}^{\dagger} J_G S S^{\dagger} S S^* J_G^*\big) \\
&= \mathrm{Tr}\big(C_{xx}^{\dagger} J_G S S^{\dagger} S^{*\dagger} S^* S S^* J_G^*\big) \\
&= \mathrm{Tr}\big(C_{xx}^{\dagger} J_G S (S^* S)^{\dagger} (S^* S) S^* J_G^*\big) \\
&= \mathrm{Tr}\big(C_{xx}^{\dagger} J_G S C_{yy}^{\dagger} C S^* J_G^*\big) \\
&= \mathrm{Tr}\big(C_{yy}^{\dagger} C_{yy} S^* J_G^* C_{xx}^{\dagger} J_G S\big) \\
&= \mathrm{Tr}\big(C_{yy}^{\dagger/2} C_{yy} S^* J_G^* C_{xx}^{\dagger} J_G S C_{yy}^{\dagger/2}\big) \\
&= \mathrm{Tr}\big(C_{yy} C_{yy}^{\dagger/2} S^* J_G^* C_{xx}^{\dagger} J_G S C_{yy}^{\dagger/2}\big) \\
&= \mathrm{Tr}\big(C_{yy} C_{yy}^{\dagger/2} C_{xy} C_{xx}^{\dagger} C_{xy}^{*} C_{yy}^{\dagger/2}\big) \\
&= \mathrm{Tr}\big(C_{yy} \mathrm{Corr}_{xy}^* \mathrm{Corr}_{xy}\big),
\end{aligned} \quad (108)$$

where, in the first equation we have used (105), in the second, third and fourth equality we have used the following standard relations

$$S = S S^{\dagger} S \qquad S^{\dagger} = S^{\dagger} S^{*\dagger} S^* \qquad (S^* S)^{\dagger} = S^{\dagger} S^{*\dagger}, \quad (109)$$

in the fifth equality we have used (101), in the eighth equality we have exploited the commuting property $C_{yy}^{\dagger/2} C_{yy} = C_{yy} C_{yy}^{\dagger/2}$, in the ninth equality we have used again (105) and, finally, in the last equality, we have introduced the definition of the correlation operator

$$\mathrm{Corr}_{xy} = C_{xx}^{\dagger/2} C_{xy} C_{yy}^{\dagger/2}, \quad (110)$$

which is used in Canonical Correlation Analysis. ∎

We now have all the ingredient for the proof of Lemma 7.

**Proof. of Lemma 7.** We start from recalling the problem we want to solve:

$$\min_{M \in \mathbb{R}^{d \times k}, b \in \mathbb{R}^d} \mathrm{Var}_{\rho}(\tau_{M,b})^2 = \min_{M \in \mathbb{R}^{d \times k}, b \in \mathbb{R}^d} \mathbb{E}_{(\mu,s) \sim \rho} \left\| w_\mu - (M\Phi(s) + b) \right\|^2. \quad (111)$$

By taking the derivatives w.r.t. $b$, we conclude that the matrix $M_\rho \in \mathbb{R}^{d \times k}$ and the vector $b_\rho \in \mathbb{R}^d$ minimizing the term above satisfy

$$w_\rho = M_\rho \, \nu_\rho + b_\rho, \quad (112)$$

or, equivalently,

$$b_\rho = w_\rho - M_\rho \, \nu_\rho. \quad (113)$$

Exploiting this equality, we can rewrite our problem above as

$$\min_{M \in \mathbb{R}^{d \times k}, b \in \mathbb{R}^d} \mathbb{E}_{(\mu,s) \sim \rho} \left\| w_\mu - (M\Phi(s) + b) \right\|^2 = \min_{M \in \mathbb{R}^{d \times k}} \mathbb{E}_{(\mu,s) \sim \rho} \left\| (w_\mu - w_\rho) - M\big(\Phi(s) - \nu_\rho\big) \right\|^2.$$

We now observe that the problem above has the same form of the problem considered in Lemma 8, once one identifies $\mathcal{X} = \mathcal{S}$ (the space of the side information), $x = s$, $y = w_\mu - w_\rho$ and $\Psi(x) = \Phi(s) - \nu_\rho$. The desired statements automatically derive from the application of Lemma 8 to our context. ∎

# D  Proofs of the statements in Sec. 4

In this section we report the proofs of the statements we used in Sec. 4 in order to prove the expected excess risk bound for Alg. 1 in Thm. 4. We start from proving in App. D.1 the properties of the surrogate functions in Prop. 3. Then, in App. D.2, we give the convergence rate of Alg. 1 on the surrogate problem in (18). We conclude by describing in App. D.3 how Alg. 1 can be implemented by computing only evaluations of the kernel associated to the feature map $\Phi$, without the need of explicitly evaluating the feature map itself. This is useful when the space in which the image of the feature map lies is high (or even infinite) dimensional.

## D.1  Proof of Prop. 3

We now prove the properties of the surrogate functions in Prop. 3.

**Proposition 3** (Properties of the surrogate meta-loss $\mathcal{L}$)**.** *For any $Z \in \mathcal{D}$ and $s \in \mathcal{S}$, the function $\mathcal{L}(\cdot, \cdot, s, Z)$ is convex, differentiable and its gradient is given by*

$$\nabla \mathcal{L}(\cdot, \cdot, s, Z)(M, b) = -\lambda \Big( A\big(\tau_{M,b}(s), Z\big) - \tau_{M,b}(s) \Big) \left( \begin{array}{c} \Phi(s) \\ 1 \end{array} \right)^{\top} \tag{19}$$

*for any $M \in \mathbb{R}^{d \times k}$ and $b \in \mathbb{R}^d$. Moreover, under Asm. 1 and Asm. 3, we have*

$$\big\| \nabla \mathcal{L}(\cdot, \cdot, s, Z)(M, b) \big\|_F^2 \leq L^2 R^2 (K^2 + 1). \tag{20}$$

**Proof.** We are interested in studying the properties of the surrogate function

$$\mathcal{L}(\cdot, \cdot, s, Z) : \mathbb{R}^{d \times k} \times \mathbb{R}^d \to \mathbb{R} \tag{114}$$

in (18). We start from observing that, such a function coincides with the composition of the Moreau envelope $\hat{\Delta}(\cdot, Z) : \mathbb{R}^d \to \mathbb{R}$ of the empirical risk $\mathcal{R}_Z$:

$$\theta \mapsto \hat{\Delta}(\theta, Z) = \min_{w \in \mathbb{R}^d} \mathcal{R}_{Z, \theta}(w) \qquad \mathcal{R}_{Z, \theta}(w) = \frac{1}{n} \sum_{i=1}^{n} \ell(\langle x_i, w \rangle, y_i) + \frac{\lambda}{2} \| w - \theta \|^2 \tag{115}$$

with the linear transformation

$$s \in \mathcal{S} \mapsto \tau_{M,b}(s) = M \Phi(s) + b \in \mathbb{R}^d. \tag{116}$$

In other words, for any $M \in \mathbb{R}^{d \times k}$ and $b \in \mathbb{R}^d$, we can write

$$\mathcal{L}(M, b, s, Z) = \hat{\Delta}(\tau_{M,b}(s), Z). \tag{117}$$

As a consequence, since the Moreau envelope is convex and differentiable [6, Prop. 12.29], the resulting surrogate function $\mathcal{L}(\cdot, \cdot, s, Z)$ is convex and differentiable over $\mathbb{R}^{d \times k} \times \mathbb{R}^d$. The closed form of the gradient in (19) directly derives from the composition rule for derivatives and the closed form of the gradient of the Moreau envelope [6, Prop. 12.29]

$$\nabla \hat{\Delta}(\cdot, Z)(\theta) = -\lambda \big( A(\theta, Z) - \theta \big) \in \mathbb{R}^d, \tag{118}$$

with $A(\theta, Z)$ defined as in (2). Consequently, we get

$$\nabla \mathcal{L}(\cdot, \cdot, s, Z)(M, b) = \nabla \hat{\Delta}(\cdot, Z)(\tau_{M,b}(s)) \left( \begin{array}{c} \Phi(s) \\ 1 \end{array} \right)^{\top}, \tag{119}$$

coinciding with the desired closed form in (19). Finally, we observe that, as shown in [13, Prop. 4], under Asm. 1, for any $\theta \in \mathbb{R}^d$, we have

$$\big\| \nabla \hat{\Delta}(\cdot, Z)(\theta) \big\|^2 \leq L^2 R^2. \tag{120}$$

As a consequence, exploiting the rewriting above, Asm. 1 and Asm. 3, we get the desired bound in (20):

$$
\begin{aligned}
\big\| \nabla \mathcal{L}(\cdot, \cdot, s, Z)(M, b) \big\|_F^2 &= \big\| \nabla \hat{\Delta}(\cdot, Z)\big(\tau_{M,b}(s)\big) \Phi(s)^{\top} \big\|_F^2 + \big\| \nabla \hat{\Delta}(\cdot, Z)\big(\tau_{M,b}(s)\big) \big\|^2 \\
&= \big\| \nabla \hat{\Delta}(\cdot, Z)\big(\tau_{M,b}(s)\big) \big\|^2 \big\| \Phi(s) \big\|^2 + \big\| \nabla \hat{\Delta}(\cdot, Z)\big(\tau_{M,b}(s)\big) \big\|^2 \\
&\leq L^2 R^2 (K^2 + 1),
\end{aligned} \tag{121}
$$

where in the second equality above we have exploited the fact that for any vectors $a \in \mathbb{R}^d$ and $b \in \mathbb{R}^s$, we have

$$\big\| a b^{\top} \big\|_F^2 = \mathrm{Tr}\big( b a^{\top} a b^{\top} \big) = \mathrm{Tr}\big( b^{\top} b a^{\top} a \big) = \| a \|^2 \| b \|^2. \tag{122}$$

∎

## D.2 Convergence rate of Alg. 1 on the surrogate problem in (18)

We now give the convergence rate of Alg. 1 on the surrogate problem in (18).

**Proposition 9** (Convergence rate on the surrogate problem in (18)). *Let $\bar{M}$ and $\bar{b}$ be the average of the iterations obtained from the application of Alg. 1 over the training data $(Z_t, s_t)_{t=1}^T$ with constant meta-step size $\gamma > 0$ and inner regularization parameter $\lambda > 0$. Then, under Asm. 1 and Asm. 3, for any $\tau_{M,b} \in \mathcal{T}_\Phi$, in expectation w.r.t. the sampling of $(Z_t, s_t)_{t=1}^T$,*

$$\mathbb{E}\,\hat{\mathcal{E}}_\rho\big(\tau_{\bar{M},\bar{b}}\big) - \hat{\mathcal{E}}_\rho\big(\tau_{M,b}\big) \leq \frac{\gamma L^2 R^2 (K^2 + 1)}{2} + \frac{\big\|(M,b)\big\|_F^2}{2\gamma T}. \tag{123}$$

**Proof.** We observe that Alg. 1 coincides with Stochastic Gradient Descent applied to the convex and Lipschitz (see Prop. 3) surrogate problem in (18):

$$\min_{M \in \mathbb{R}^{d \times k}, b \in \mathbb{R}^d} \hat{\mathcal{E}}_\rho(\tau_{M,b}) \qquad \hat{\mathcal{E}}_\rho(\tau_{M,b}) = \mathbb{E}_{(\mu,s) \sim \rho}\, \mathbb{E}_{Z \sim \mu^n}\, \mathcal{L}\big(M, b, s, Z\big). \tag{124}$$

As a consequence, by standard arguments (see e.g. [36, Lemma 14.1, Thm. 14.8] and references therein), for any $\tau_{M,b} \in \mathcal{T}_\Phi$, we have

$$\mathbb{E}\,\hat{\mathcal{E}}_\rho\big(\tau_{\bar{M},\bar{b}}\big) - \hat{\mathcal{E}}_\rho\big(\tau_{M,b}\big) \leq \frac{\gamma}{2T} \sum_{t=1}^T \mathbb{E}\,\big\|\nabla\mathcal{L}\big(\cdot,\cdot,s,Z_t\big)(M_t, b_t)\big\|_F^2 + \frac{\big\|(M,b)\big\|_F^2}{2\gamma T}. \tag{125}$$

The desired statement derives from combining this bound with the bound on the norm of the meta-subgradients in (20) in Prop. 3. $\blacksquare$

## D.3 Implementation of Alg. 1 with kernels

We conclude this section by describing how Alg. 1 can be implemented by computing only evaluations of the kernel associated to the feature map $\Phi$. We describe this in the following lemma exploiting standard arguments from online learning with kernels literature (see e.g. [25, 36, 37]).

**Lemma 10** (Implementation of Alg. 1 by kernel's evaluations). *Let $\big(M_t, b_t, \theta_t\big)_{t=1}^T$ be the iteration generated by Alg. 1 with meta-step size $\gamma \geq 0$. Then,*

$$\theta_{t+1} = -\gamma \sum_{j=1}^t \nabla\hat{\Delta}(\cdot, Z_j)(\tau_{M_j, b_j}(s_j))\, k(s_j, s_{t+1}) + b_{t+1}, \tag{126}$$

*where the function $\hat{\Delta}$ and its gradients $\nabla\hat{\Delta}(\cdot, Z_j)$ are defined in (115) and (118) above and we have introduced the evaluation*

$$k(s_j, s_{t+1}) = \Phi(s_j)^\top \Phi(s_{t+1}), \tag{127}$$

*of the kernel associated to the feature map $\Phi$.*

**Proof.** Exploiting the closed form of the meta-subgradient in (19) in Prop. 3, we can rewrite more explicitly the update step of Alg. 1 as follows:

$$\begin{aligned}
M_{t+1} &= M_t - \gamma\,\nabla\hat{\Delta}(\cdot, Z_t)(\tau_{M_t, b_t}(s_t))\Phi(s_t)^\top \\
b_{t+1} &= b_t - \gamma\,\nabla\hat{\Delta}(\cdot, Z_t)(\tau_{M_t, b_t}(s_t)) \\
\theta_{t+1} &= M_{t+1}\Phi(s_{t+1}) + b_{t+1}.
\end{aligned} \tag{128}$$

By induction argument on the iteration $t$, one can easily see that the update of the matrix $M_{t+1}$ can be equivalently rewritten as

$$M_{t+1} = -\gamma \sum_{j=1}^t \nabla\hat{\Delta}(\cdot, Z_j)(\tau_{M_j, b_j}(s_j))\Phi(s_j)^\top. \tag{129}$$

As a consequence, we can rewrite the update of the bias vector $\theta_{t+1}$ as follows

$$
\begin{aligned}
\theta_{t+1} &= M_{t+1}\Phi(s_{t+1}) + b_{t+1} \\
&= -\gamma \sum_{j=1}^{t} \nabla\hat{\Delta}(\cdot, Z_j)(\tau_{M_j, b_j}(s_j))\Phi(s_j)^{\top}\Phi(s_{t+1}) + b_{t+1} \\
&= -\gamma \sum_{j=1}^{t} \nabla\hat{\Delta}(\cdot, Z_j)(\tau_{M_j, b_j}(s_j))\, k(s_j, s_{t+1}) + b_{t+1}.
\end{aligned}
\tag{130}
$$

This last equation coincides with the desired statement. ∎

## E  Experimental details

In this section we report the implementation details we omitted in the main body.

In order to tune the hyper-parameters $\lambda$ and $\gamma$ our experiments, we followed the same validation procedure described in [13, App. I]. Such a procedure requires performing a meta-training, a meta-validation and a meta-test phase on a separate sets of $T_{\text{tr}}$ training tasks, $T_{\text{va}}$ validation tasks and $T_{\text{te}}$ test tasks. Each task in the training set is observed by a corresponding dataset $Z_{\text{tr}}$ of $n = n_{\text{tr}}$ points, while, the tasks in the test and validation sets are all provided with a corresponding training dataset $Z_{\text{tr}}$ of $n_{\text{tr}}$ points and a corresponding test dataset $Z_{\text{te}}$ of $n_{\text{te}}$ points.

Specifically, in our experiments, we applied the validation procedure above as described in the following.

**Synthetic clusters.** We considered 14 candidates values for both $\lambda$ and $\eta$ in the range $[10^{-5}, 10^5]$ with logarithmic spacing and we evaluated the performance of the estimated meta-parameters (bias vectors) by using $T = T_{\text{tr}} = 300$, $T_{\text{va}} = 100$, $T_{\text{te}} = 80$ of the available tasks for meta-training, meta-validation and meta-testing, respectively. In order to train and to test the inner algorithm, we splitted each within-task dataset into $n = n_{\text{tr}} = 50\% \, n_{\text{tot}}$ for training and $n_{\text{te}} = 50\% \, n_{\text{tot}}$ for test.

**Synthetic circle.** We considered 16 candidates values for both $\lambda$ and $\eta$ in the range $[10^{-7}, 10^7]$ with logarithmic spacing and we splitted the data as in the clusters' settings above.

**Lenk dataset.** We considered 14 candidates values for both $\lambda$ and $\eta$ in the range $[10^{-5}, 10^5]$ with logarithmic spacing and we evaluated the performance of the estimated meta-parameters by splitting the tasks into $T = T_{\text{tr}} = 100$, $T_{\text{va}} = 40$, $T_{\text{te}} = 30$ tasks used for meta-training, meta-validation and meta-testing, respectively. In order to train and to test the inner algorithm, we splitted each within-task dataset into $n = n_{\text{tr}} = 16$ for training and $n_{\text{te}} = 4$ for test.

**Schools dataset.** We considered 14 candidates values for both $\lambda$ and $\eta$ in the range $[10^{-5}, 10^5]$ with logarithmic spacing and we evaluated the performance of the estimated meta-parameters by splitting the tasks into $T = T_{\text{tr}} = 70$, $T_{\text{va}} = 39$, $T_{\text{te}} = 30$ tasks used for meta-training, meta-validation and meta-testing, respectively. In order to train and to test the inner algorithm, we splitted each within-task dataset into $n = n_{\text{tr}} = 75\% \, n_{\text{tot}}$ for training and $n_{\text{te}} = 25\% \, n_{\text{tot}}$ for test.

We conclude this section reporting the characteristics of the machine we used for running our experiments and the complexity of our method in Alg. 1.

All the experiments were conducted on a workstation with 4 Intel Xeon E5-2697 V3 2.60Ghz CPUs and 256GB RAM.

The variant of our method in Alg. 1 for biased regularization using the batch inner algorithm in (2) has a time and space complexity $\mathcal{O}(d(k + n))$. The variant for fine tuning using the online inner algorithm in (3) has a time and space complexity $\mathcal{O}(dk)$.