[Reviews · NeurIPS 2020]

Review 1

Summary and Contributions: The authors propose a new setting for meta-learning, one where there exist additional information about a task at hand, which can be utilized to better disentangle the tasks in the case where the tasks are sourced from a heterogenous task distribution. More specifically, the authors argue that if the tasks are different enough that they require different initial conditions for a neural network to perform well at them, then generating an initialization or parameterization conditional on externally available task information should improve the performance. The intuition is solid, and the theoretical and empirical evaluations seem to clearly indicate that this is the case.

Strengths: There already exists a body of research papers that conditions initial conditions or parameterization upon task information (usually using a task-level embedding that generates weights, using a form of a hyper network). However, this work sets some theoretical grounding for the effectiveness of such methods vs methods with static initializations/parameterizations for the base-model. I consider this paper a complementary piece of work that provides some further perspectives on why such methods might be a good idea in heterogeneous task distributions.

Weaknesses: It's weak novelty-wise, and demonstrates the effectiveness of a task-conditional parameterizations that has already shown promise in other works. Moreover, for the proposed setting to work, task side-information is required, and the fact that with more data you can build a better model is very intuitive and tends to generalize across the deep learning landscape. Yes. It's cool to see some theory about why this is, as well as some synthetic experiments that showcase this, but it's neither the first method to propose hyper-net style parameter generation, nor the first to show that having more task information can help improve generalization.

Correctness: The claims and evaluation appear correct.

Clarity: The writing quality needs improvement. Please see my extra comments section. If the authors can address this concern, I'd be happy to upgrade my score as a weak accept.

Relation to Prior Work: The authors make a fair effort to relate to some previous work, but not elaborate enough in my opinion.

Reproducibility: Yes

Additional Feedback: Abstract: Sentence 1: Intro to existing approaches for meta-learning in few-shot learning. However, this sentence does feel a bit problematic. Fine tuning is simply an adaptation method, and biased regularization is a regularizer for meta-learning. I’d say at best, these are building blocks for meta-learning algorithms. Not full learning algorithms. Sentence 2: More details on the previously introduced methods. Sentence 3: Limitations of those methods. Good. Sentence 4: Proposed method. Good. Sentence 5: Environments under which the proposed method can be useful. Good. Sentence 6: A practical variant of the proposed method. Good. Sentence 7: Experiments. This is missing key results. The abstract suffers from clarity and readability problems. An example: “They have been shown to be effective to tackle distributions of tasks, in which the tasks’ target vectors are all close to a common meta-parameter vector.” What is a task’s target vector? Did you mean that ‘good parameters for a target task’ are close to a common parameter setup that can be meta-learned? Correct me if I am wrong. But if I am right, then the writing needs to offer additional clarity to the reader, otherwise it makes understanding the work very hard, and requires the reader to actually be very well versed in the subject to ‘guess’ what is meant. Overall I think the idea itself is very interesting. What I like to call ‘dynamic parameter initializations’. Similar to work by Andrei A. Rusu et al in Latent Embedding Optimization. Introduction: Paragraph 1: Intro to a number of meta-learning methods. Paragraph 2: Limitation of those meta-learning methods. Contributions and organization: This was a helpful paragraph. Overall the introduction lacks information on what the problem that we care about actually is. Yes. Meta-learning methods are introduced, but meta-learning itself isn’t. There isn’t much motivation on why it is worthwhile to study meta-learning either. You do note a gap and propose a method to address it. You fail to explain how your method differs from the other conditional meta-learning methods.


Review 2

Summary and Contributions: This paper proposes and conducts theoretical analysis on a meta-learning algorithm that learns a mapping from side information to provide initialization for the inner gradient descent algorithm. It analyzes its excess risk measured by the conditional variance of the optimal task solution, and shows its advantage to the meta-learning algrithms with a constant initialization theoretically and in a few simple datasets.

Strengths: It conducts rigorous analysis on the statistical efficiency of the proposed conditional meta-learning algorithm and compares it to unconditional meta-learning and independent task learning. The propositions and theorems in the main text seem correct although I have not checked all the proofs in the supplementary.

Weaknesses: The proposed method is restricted to the family of linear regression models in the inner loop and linear conditional mapping with the side information. While the resulting objective is convex, it is far from actual meta-learning algorithms that would be used in practice. I wonder how useful it will be for real-world application. The choice of the hyper-parameters \lambda and \gamma depend on unknown quantities. In experiment, the authors use a validation task set to choose those values. Could you comment on how it will impact the excess risk? The proposed method requires the side information to be conditional independent from the training data. When the side information is a different set of data points, one may choose to augment the training data with the side information and estimate w^* with a constant initialization. It is not clear to me if the proposed algorithm will make better use of the side information than this naive approach. If so, is there an optimal spit of the dataset between training data size and the side information data size?

Correctness: The proof appears to be correct but I haven't checked the proofs in the supplementary. In (25), is it the comparison between proposed conditioning function vs unconditioning meta-learning or between unconditioning meta-learning vs optimal conditioning function? If it is the latter, under what condition will the proposed conditioning method be better than the unconditoning method? The same question for (26).

Clarity: The algorithm is clearly presented and analyzed.

Relation to Prior Work: This paper lists prior works that analysize meta-learning algorithms using task information and claims this is the first work that does theoretical investigation with a convex formulation. However, I'm not familiar with the literature of applying learning theory to meta-learning problems.

Reproducibility: Yes

Additional Feedback:


Review 3

Summary and Contributions: The limitation of existing meta-learning works is that they might fail to capture the complex distribution of heterogeneous tasks due to their strong assumption that one meta-parameter vector encodes task distribution. To overcome this limitation, this work proposes to use the task’s side information as a condition of meta-parameter vector and theoretically demonstrates the advantage of the conditional meta-learning.

Strengths: In recent years, despite the explosive publication of meta-learning papers since many researchers focused on the importance of meta-learning, meta-learning research rarely handles real-world scenarios effectively. I think one of main reason is most of the published researches only focus on homogeneous tasks. From this point of view, the message of this paper that conditions such as side information is important to handle heterogeneous tasks is timely appropriate and useful.

Weaknesses: However, another limitation of the existing meta-learning works is to lack of scalability to apply real-world scenarios. To overcome those limitations toward real-world applications, recent meta-learning papers, Meta-dataset[1] and Bayesian TAML[2], provide scalable heterogeneous tasks (Meta-dataset and Multi-dataset, respectively) and propose meta-learning models successfully working well on those large-scale heterogeneous environments. I think the direction of these days meta-learning researches should include applicability to real-world scenarios to some extent outside the lab environment. Therefore, I highly recommend this work also demonstrates the effectiveness of their analysis by applying Algorithm 1 on the heterogeneous tasks of Meta-dataset and Multi-dataset to support the theoretical analysis. [1] Triantafillou, Eleni, et al. "Meta-dataset: A dataset of datasets for learning to learn from few examples." The International Conference on Learning Representations. 2020. [2] Lee, Hae Beom, et al. "Learning to Balance: Bayesian Meta-Learning for Imbalanced and Out-of-distribution Tasks." The International Conference on Learning Representations. 2020.

Correctness: Their claims are theoretically well proved and they empirically prove it on simple Synthetic clusters and Synthetic circle experiments. However, I think this claim would be validated on the meta-learning benchmark dataset (Meta-dataset[1]) consisting of heterogeneous tasks. [1] Triantafillou, Eleni, et al. "Meta-dataset: A dataset of datasets for learning to learn from few examples." The International Conference on Learning Representations. 2020.

Clarity: Yes. This paper is clearly written and easy to follow.

Relation to Prior Work: In section 2 and 3, this work describe the difference between existing unconditional meta-learning and this work. However, I think the section which how the theoretical analysis of this work interpret the existing conditional meta-learning works to tackle heterogeneous tasks would be added.

Reproducibility: Yes

Additional Feedback:


Review 4

Summary and Contributions: The authors consider the simultaneous learning of several linear prediction functions for tasks whose relation to each other is characterized by side information expressed by its own feature representation. The setting is typical, for example, in recommendation systems where learning tasks are different customers and products to be recommended for the customers correspond to the data so that both the customers and products would have their own feature representation. The authors compare three approaches, the first of which solves the tasks independently and by ignoring the side information so that a learning algorithm infers the weight vectors for the tasks by minimizing the loss on the training data plus a standard quadratic regularizer. The second approach, unconditional meta-learning, solves the tasks again without the side information but jointly so that the regularizer encourages the weight vectors for different tasks to be close to a jointly inferred bias vector rather than close to zero vector. The third approach, conditional meta-learning, infers a linear mapping from the side information feature space to the data feature space such that the regularized encourages weight vectors close to the mapped side information.

Strengths: The manuscript presents a rigorous theoretical analysis of the properties and benefits of the considered learning approaches. The analysis show the excess risk of the conditional approach to decrease relatively to the square root of the number of tasks and that it avoids the negative transfer effect from which meta or transfer learning approaches often tend to suffer. Experimental evidence on the benefits of meta-learning are shown with synthetic data and in the appendix also with real world tasks.

Weaknesses: [After rebuttal] I have read the rebuttal and on my behalf the reviewers' concerns have been addressed. [Original review] As a potential weakness, the reviewer could think that the consideration appears somewhat isolated from the other machine learning approaches usable for similar situations. The focus of the manuscript is of course on showing the benefits of taking advantage of the side information in meta learning with linear methods and at the same time considering other learning approaches has the risk on obfuscating the message. Nevertheless, it could cross the mind of the reader whether the proposed approach is worth considering if the performance is not compared with other approaches.

Correctness: The claims and method appear to be correct as far as the reviewer can see. Due to the short time and huge review burden, careful review of the proofs in the appendix was not possible.

Clarity: The paper is very clearly written and the mathematics is elegant.

Relation to Prior Work: Prior work and the contribution over it is clearly discussed, though, the reviewer may not be too aware of the recent literature on the topic.

Reproducibility: Yes

Additional Feedback:


Review 5

Summary and Contributions: This paper studies the sample complexity of (conditional) meta-learning. Recent prior works on online meta-learning, have assumed there is a single global meta-parameter close to each target tasks parameter vector. In the task/data generative model in this work the key modeling difference is that there is additional side information $s$, which when conditioned on provides information about the underlying task. Hence instead of fitting a single global regularization parameter (which is essentially an average across the tasks), this paper allows learning of a task-conditioned function $\tau(s)$ in the regularizer--which allows finer grained adaptation to each task. Some motivation could be provided to further motivate the relevance/practicality of this set-up. Several theorems/examples are proved in the generic setting where one assumes access to an ERM oracle (i.e. Theorem 1) to highlight the benefits of conditional meta-learning vs unconditional meta-learning. The key point being one pays an effective variance wrt. \tau (Eq .7) which captures how well the function \tau(s) can capture each task mean. Example 1 illustrates in a GMM/regression setting how side info in the form of covariates can improve over unconditional info. Sec. 4 provides a more practical SGD-style meta-algorithm which restricts \tau to be a linear map and shows a SA-style guarantee when applied to a convex surrogate risk. Eq. 24 highlights important theoretical features of the guarantee. Several synthetic expts are then shown in the main (and some real data experiments in the Appendix) to highlight the utility of the method.

Strengths: Moving beyond the “single” global task parameter assumption in recent work seems an important research direction, as this setting is important but certainly limited in its descriptive power. The primary contribution of this paper is the modeling set-up for the (conditional) side information variant of meta-learning. The bounds provided are intuitive and naturally capture the conditioning effect which is a byproduct of the side-information--this is nicely illustrated in a concrete setting in Example 1 with a cluster of tasks, where one can see how the cluster distances/sample sizes/task number factor into a reasonable problem. Similarly, the SGD-style algorithms bound in Eq. 24 nicely highlights the trade-off between task-number (in suppressing the complexity of the \tau(s) class) and the optimal conditional variance in hindsight. Moreover, the assumptions/algorithms seem natural (in the setting where the dataset is used as the conditioning information -- see also later) and the method seems practical/shows gains in situations where they might be expected. The bounds also “back off” to unconditional guarantees under appropriate parameter setting.

Weaknesses: The bounds naturally capture the risk-reduction due to the conditional variance--that being said the stability techniques/decompositions used in the bounds are not entirely surprising (and do not seem to encounter significant technical difficulties); although this is not perhaps a significant as the modeling set-up seems to be the key contribution. The best bounds rely on optimal choices of regularization parameters; i.e. Theorem 1 requires the conditioning function to depend on variance with respect to some \tau which factors into the variance upper bound; as this variance upper bound relies on population quantities it is not clear a good choice of \tau can be selected apriori. The bound for practical SGD algorithm has a similar flavor where upon assuming realizability; the algorithm depends on $M, b$ so obtaining the “oracle” performance requires having a handle on the optimal variance. This often occurs in regularized bounds though and in experimental situations it appears this issue can be mitigated with cross-validation. The abstraction of the framework in terms of the task-sampling model/guarantees are nice--but the introduction of “side information” initially comes across as somewhat unmotivated. Useful references are indeed given, although to my knowledge [20] for example learns the mixture components using an EM-style algorithm in a completely unsupervised way (so no “extra” information is needed). In the actual examples this framework is instantiated by using extra covariates $x_i$ as side-information; in experiments the original data is used which this alluded to in Remark 2 but could be explained more clearly. It seems in general this requires either data-splitting or actual side-information. The concrete cases in Example 1 seem most related to practice and here I believe it's the case that the (x, y) data is Z and the extra covariates s=(x_i) should be interpreted in the sense of a semi-supervised set-up. Similarly in the synthetic/real experiments it's not clear to me what the side information is (i.e. collections of (x,y) that are extra or using sample splitting on the total dataset)--as using sample splitting can also lead to reduced practical performance. In the synthetic circle example the situation is more clear in reference to the above; although this example seems somewhat more contrived. I believe the motivation for the modeling assumptions/explanation of these specific instantiations is important as it is one of the key contributions of the work; especially to see under which concrete generative models theoretical improvement is provided.

Correctness: The claims seem correct.

Clarity: The paper is overall well-written and coherent (aside from a few small typos); although I think the specific instantiations of the way side information is used could benefit from further motivation/explanation.

Relation to Prior Work: Seems well-cited and compared too.

Reproducibility: Yes

Additional Feedback:


Review 6

Summary and Contributions: This work formulates the problem of conditional meta-learning using the notion of side information for tasks. Side information can be mapped to parameters that are either used as initialization for fine-tuning or for biased regularization. Excess risk bounds are shown for this setting along with intuitive meta-learning examples that demonstrate how conditional meta-learning can be better than unconditional meta-learning. An efficient online meta-learning algorithm is provided for a convex formulation of this setting, with excess risk guarantees that depend on number of train tasks and samples per task. Experiments are conducted for some simulations and real world datasets to show the benefits of this algorithm.

Strengths: - Problem formulation: Given the growing interest in conditional meta-learning to learn task specific initializations, the clean and sound mathematical formulation (for biased regularization and fine-tuning) from this work can serve as a good starting point for subsequent theoretical work on this. - Theoretical analysis: The paper provides good intuitions for the theoretical results in the main paper, backed by complete proofs in the appendix. The example of clustered tasks in Example 1 also highlights how an extra (unlabeled) dataset as side information can potentially help with sample efficiency of new task, and how the similarity within clusters and separation between clusters can impact the gap. - Experiments: This work demonstrate that even the simple convex formulation, where the task specific initialization is an affine function of fixed features of side information, can show practical improvements over a common learned initialization (at least for the simple datasets they consider).

Weaknesses: - While interesting, the theoretical setting and analysis are both fairly straightforward extensions of the unconditional setting in [13] (citation from paper) The main differences are the introduction of a different variances term (which shows up naturally by independence of s and Z conditioned on \mu), convexity of the surrogate problem (that follows from the linear parameterization composed over the Moreau envelope) and norm bounds (by assumptions). It would be nice to have a short summary of the changes in analysis for clarity and full disclosure. - There is not enough discussion about how to place this work in context of prior work on conditional meta-learning. (more details below) Despite its simplicity, by virtue of being one of the first theoretical analyses for the unconditional setting, I believe that the overall contribution of this paper is positive.

Correctness: The proofs mostly use standard tools and look correct. The empirical methodology seems mostly correct, though some small details are missing (detailed comments below)

Clarity: - Theoretical results: This part is fairly clearly written. The complexity of notation in places is perhaps unavoidable. ---- The notation R^{\lambda}_Z in equation (2) is confusing, since the definition involves \theta but notation doesn't. Perhaps something like R^{\lambda,\theta}_Z is better. - Experimental setup: This section could use some restructuring. ---- Details about how the side information is used in the experiments (including the features \phi) is crucial and is better placed in the main paper rather than appendix. ---- The experiments on real datasets can also be moved to main paper. The circle experiment is not very insightful in my opinion and can be deferred to the appendix instead. ---- It is not clear how much data per task is used as side info and training data respectively for all of the conditional experiments.

Relation to Prior Work: - It would help to have a short description of some of the conditional meta-learning approaches proposed in prior work, including a discussion about which ones are the most similar to this setting and whether a similar strategy of linear function of features for finetuning/biased regularization has been considered before. - There seem to be a theoretical result for conditional meta-learning in [38] (citation from paper). It would be good to have a discussion/comparison with the results from [38].

Reproducibility: Yes

Additional Feedback: - All examples and experiments (except for the circle example) use an additional dataset as side information. Thus it would help to have a more detailed discussion about this setting beyond just Remark 2. It is not clear (not enough examples are provided) as to what other kinds of side info can be, apart from an additional dataset. The circle example is one such instance, albeit a very artificial one. - It appears that only additional *unlabeled data* is needed as side info in all cases considered (Example 1 and experiments). This is worth pointing out, since it first seemed to me that the additional side info dataset could instead be use by an unconditional method as additional samples. However this is not the case if side info is unlabeled. Also worth mentioning that the clustering of tasks in this case will be characterized by the similarity in data distributions for tasks (as also in Example 1). - Is the featurization of a dataset (as mean of \phi(x) from Section E.2) universal for universal features \phi? In other words, can you always express \tau_\rho(D) using this kind of a representation or can you lose out on expressivity?

[Author Response · NeurIPS 2020]

We thank the reviewers for the feedback. We reply to the outstanding points below.

**REVIEWER 1.**

R. *It's weak novelty-wise, the effectiveness of task-conditional parameterizations have already shown promise in other works.* A. We are not aware of other papers (nor the reviewer is providing any reference) proposing an efficient meta-learning method addressing heterogeneous environments of tasks which is computationally efficient (i.e. based on a convex method) and supported by learning guarantees. Note that the method we propose and analyze is based on similar ideas already used in literature, but it is a new method designed by us. All the related works we know are cited in the paper, but we are happy to discuss any other related paper we missed. R. *The abstract suffers from clarity problems.* A. Note that we described fine tuning and biased regularization as meta-learning strategies (not algorithms). As conventional in literature, we denoted by task's target vector the minimum norm weight vector minimizing the true risk associated with that task. We will clarify this. R. *In the introduction there isn't much motivation on meta-learning.* A. We decided to briefly recall the well-known intuition on which standard meta-learning is based on in order to mainly focus on the less known conditional setting, the main topic of the paper. R. *You fail to explain how your method differs from the other conditional meta-learning methods.* As explained in the paper, the related works we know consider different formulations of the problem and they do not provide a complete theoretical analysis for the methods they propose. Because of this, drawing a complete comparison between our method and the preexisting literature is not always easy. We will add more details about this comparison where possible.

**REVIEWER 2.**

R. *The proposed method is restricted to the family of linear regression models.* A. Note that a feature map can be also added in the within-task problem. In our experimental settings and in many other scenarios in literature (such as [5,13,14,22] in our paper), the use of linear models with the right feature map revealed to be sufficiently effective. Extending our method and the corresponding analysis to the non-convex setting is for sure an interesting direction we leave for future research. R. *The choice of the hyper-parameters $\lambda$ and $\gamma$ depends on unknown quantities.* A. In practice, we validated the hyper-parameters. We are confident about the fact that a more efficient parameter-free version of our method can be developed by using arguments based on coin betting algorithms (see [12, 30] in our paper). We will investigate this in the future. This however would not change the main message that we want to convey in the paper: developing a meta-learning method for heterogeneous tasks and providing theoretical guarantees for it. R. *One may choose to augment the training data with the side information and use a constant initialization. Is there an optimal split?* A. In experiments we observed that even though we augmented the training datapoints, the standard unconditional approach performed worse than the conditional one. We guess that the independence between the side information and the training dataset can be removed, by using alternative generalization arguments, but this requires further investigation. The optimal splitting size is an interesting point, still open. R. *Regarding Eq. (25) and (26).* A. Eq. (25) and Eq. (26) are the state-of-the-art bounds for optimal unconditional meta-learning and independent task learning (ITL). Both these bounds are deduced from the general bound in Thm. 4 for our conditional meta-learning approach. This means that our method includes also unconditional meta-learning and ITL. We investigate the advantage of using the conditional approach with respect to the unconditional one and ITL in Sec. 3.

**REVIEWER 5.**

R. *A limitation of the existing meta-learning is the lack of scalability to apply real-world scenarios.* A. As described in the last appendix with experimental details, our fine tuning method scales linearly with the dimension of the features space and the dimension of the image space of the side information's feature map. To develop even more scalable methods is for sure an interesting future direction, but, for the moment, the priority was to develop a theoretically grounded method, something which is missing in literature. R. *About relation to prior work.* A. See reply to reviewer 1.

**REVIEWER 6.**

R. *It could cross the mind of the reader whether the proposed approach is worth considering if the performance is not compared with other approaches.* A. Please note that the main question we addressed in this work was to understand when the conditional meta-learning approach results to be advantageous with respect to the standard unconditional one and independent task learning. We thus think that adding to the comparison other methods would complicate the message of the paper.

[Meta-Review · NeurIPS 2020]

Thank you for bringing up the shortcomings of the initial reviewers. I'm disappointed that they did not seem to evaluate the core contributions of the paper, which are theoretical in nature. [And while I agree with some of the sentiments of the reviewers, this paper isn't setting out to solve the problems that the reviewers raise.] After the rebuttal, I sought out and found two emergency reviewers to the paper who are better suited to review this paper. The two new reviewers (whose reviews should be visible) both scored the paper above the bar. I generally agree with their assessment, as well as their feedback on the paper. I encourage the authors to incorporate their valuable feedback into the camera-ready version of the paper, including: - better motivation for the use of side information - more discussion of this work in relation to prior conditional meta-learning works - suggested adjustments of the notation - moving experiments on real datasets & other experimental details to the main text (which should be possible with the extra page) - ideally a more realistic experiment where the side information is not data Beyond these reviewer's comments, I have two additional pieces of feedback: 1. It would be valuable to discuss the connection between this work and the theoretical findings of [A], since the motivation of conditioning is somewhat at odds with the result in [A] that unconditional meta-learning is maximally expressive given a large enough network. 2. A lot of the terminology in this paper diverges from the terminology used in other meta-learning papers, especially more empirically-focused papers. For example, there is "biased regularization and fine-tuning" versus "gradient-based meta-learning"/"optimization-based meta-learning", and "conditional meta-learning" vs. the names of prior methods like LEO and multi-modal MAML. It would be helpful to draw more of a bridge between the two sets of terminology in the text of the paper, to better help readers connect papers and methods in the field of meta-learning. [A] Finn & Levine. Meta-Learning and Universality: Deep Representations and Gradient Descent can Approximate any Learning Algorithm. ICLR '18